

# Formation and composition of organic aerosols from the uptake of glyoxal on natural mineral dust aerosols: a laboratory study

Francesco Battaglia[1], Paola Formenti[1], Chiara Giorio[2], Mathieu Cazaunau[3], Edouard Pangui[3], Antonin Bergé[3], Aline Gratien[1], Thomas Bertin[3], Joel F. de Brito[4], Manolis N. Romanias[4], Vincent Michoud[1], Clarissa Baldo[1,3], Servanne Chevaillier[3], Gael Noyalet[3], Philippe Decorse[5], Bénédicte Picquet-Varrault[3], and Jean-François Doussin[3]

[1] Université Paris Cité and Univ Paris Est Créteil, CNRS, LISA, F-75013 Paris, France
[2] Yusuf Hamied Department of Chemistry, University of Cambridge, Lensfield Road, Cambridge, CB2 1EW, UK
[3] Univ Paris Est Créteil and Université Paris Cité, CNRS, LISA, F-94010 Créteil, France
[4] IMT Nord Europe, Institut Mines-Télécom, Université de Lille, Centre for Energy and Environment, 59000, Lille, France
[5] Université Paris Cité, CNRS, Itodys, F-75013 Paris, France

Corresponding author: P. Formenti (paola.formenti@lisa.ipsl.fr)

## Abstract

The uptake of glyoxal on realistic submicron mineral dust aerosol particles from a natural soil (Gobi Desert) is investigated during experiments in a large simulation chamber, under variable experimental conditions of relative humidity, irradiation, and ozone concentrations. The uptake of glyoxal on the dust particles starts as soon as the glyoxal is injected in the chamber. At 80% RH, the measured uptake coefficient of glyoxal on mineral dust is $\gamma = (9 \pm 5) \times 10^{-3}$. The totality of the mass of reacting glyoxal is transformed in organic matter on the surface of the dust particles. The uptake of glyoxal is accompanied by the appearance marker peaks in the organic mass spectra and a persistent growth in the volume concentration of the dust particles. While the mass of the organic matter on the dust rapidly reverts to values prior to uptake, the organic composition of the dust is modified irreversibly. Glycolic and other organic acids but also oligomers are detected on the dust. At 80% RH, compounds ranging from $C_4$ to $C_{10}$ are observed as oligomerization products of glyoxal mono- and di-hydrate forms. The study suggests that dust aerosols could play a very substantial role in the formation of organic aerosols at high relative humidity, but also that the reaction could have potential important implications for the dust optical and hygroscopic properties, including their pH.



## 1 Introduction

Mineral dust originates naturally from the wind erosion of arid or semi-arid soils, resulting in the suspension of particles with diameters from fractions to hundreds of microns, which can be transported over thousands of kilometres whilst in the atmosphere (Adebiyi et al., 2023; Mahowald et al., 2014). The total global mass of mineral dust particles emitted annually in the atmosphere is of the order of 4600 Tg $yr^{-1}$, accounting for approximately 40% of the total annual aerosol emissions (Knippertz and Stuut, 2014; Kok et al., 2021). Major natural source areas of mineral dust are North Africa (~50% of the global annual dust emissions), Asia (~40%), North America, and the Southern Hemisphere (~10%; Kok et al., 2023). Anthropogenic emissions are associated with soil erosion for agriculture, pasture, and deforestation (Tegen and Fung, 1995; Webb and Pierre, 2018), but their contribution to the total annual dust mass loading is uncertain, ranging from 5 to 60% (Chen et al., 2023). Mineral dust significantly impacts the Earth's energy balance by absorbing and scattering radiation in the solar and terrestrial spectra (Di Biagio et al., 2019; Kok et al., 2023) and by influencing the lifetime and optical properties of mixed-phase and ice clouds (e.g., Atkinson et al., 2013; Harrison et al., 2001; Steinke et al., 2016). Current estimates of the effective radiative forcing (sum of direct and indirect) of natural mineral dust are in the range of -0.07 ± 0.18 W $m^{-2}$ (Kok et al., 2023), owing to large uncertainties in the atmospheric mass loading and properties of dust at emission and during transport (Castellanos et al., 2024; Li et al., 2021).

Gas-particle interactions along the dust lifecycle contribute to these uncertainties. Numerous laboratory and field studies show that mineral dust is capable to adsorb various reactive gaseous compounds, which may modify its chemical composition, and in turn to alter optical properties, hygroscopicity and ice nucleation activity but also may affect the oxidative capacity of the atmosphere (Bauer et al., 2007; Chirizzi, 2017; Crowley et al., 2010; Joshi et al., 2017; Liu et al., 2013; Ooki and Uematsu, 2005; Romanias et al., 2012; Seisel et al., 2004; Tang et al., 2017; Turpin and Huntzicker, 1995; Usher et al., 2003; Wagner et al., 2008). Dust aerosol may promote photocatalytic reactions of inorganic gases such as $SO_2$ and $NO_2$, initiating nucleation events (Dupart et al., 2012; Nie et al., 2014).

The uptake of volatile organic compounds (VOCs) on mineral dust such as limonene, toluene (Romanías et al., 2016), isoprene (Zeineddine et al., 2017), phenol




(Hettiarachchi and Grassian, 2024), and dicarboxylic acids (Ponczek et al., 2019), is
also documented. These reactions may alter the VOC budget in the atmosphere and
lead to the formation of secondary organic aerosols (Li et al., 2019; Tang et al., 2017;
Usher et al., 2003; Xu et al., 2023; Zeineddine et al., 2023), one of the key player of
atmospheric chemistry (Shrivastava et al., 2017).
Glyoxal (CHOCHO) is one of the most abundant VOCs in the troposphere (Lewis et
al., 2020). Atmospheric glyoxal is produced through the oxidation of aromatic
compounds like benzene, toluene, and p-xylene (Volkamer et al., 2001) as well as by
the photochemical oxidation of isoprene (Chan et al., 2017). The global atmospheric
concentrations have been evaluated in the range of 10 – 100 pptv by Fu et al. (2008).
However, case studies show sometimes higher concentrations. During a field study in
Shanghai in the summer of 2018, Guo et al. (2021) reported an average glyoxal
concentration of 164 ± 73 pptv, due to daytime photochemistry. Local concentrations
of up to 400 pptv have been documented in regions influenced by aromatic pollution
(Li et al., 2022). Satellite measurements of glyoxal show that the highest
concentrations in tropical and sub-tropical regions are found during warm, dry periods
influenced by biogenic emissions and vegetation fires, but also anthropogenic pollution
(Vrekoussis et al., 2009). Elevated glyoxal concentrations have been observed in aged
biomass burning plumes and tropical ocean regions, revealing model under-
predictions in high-emission areas due to missing complex organic compound sources
(Kluge et al., 2023). Field measurements in the north east Atlantic Ocean reveal that
models generally underestimate glyoxal concentrations due to missing contributions
from acetaldehyde and other chemical precursors, and a potential glyoxal source from
the ocean surface organic microlayer, particularly significant at night (Walker et al.,

97 2022).

Glyoxal is a very soluble molecule which readily oligomerises in water, leading to the
formation of larger molecules (Kalberer et al., 2004; Shapiro et al., 2009). It also has
the ability to uptake onto aerosol particles, potentially serving as a significant source
of organic aerosols (e.g., Liggio et al., 2005b, Carlton et al., 2007; Ervens and
Volkamer, 2010; Galloway et al., 2009; Knote et al., 2014). The uptake of glyoxal on
ammonium sulphate particles can lead to the formation of carbon-nitrogen compounds
(such as imidazole derivatives), oligomers, and organic acids (Galloway et al., 2009),
that has been observed to cause their browning (De Haan et al., 2020). The light-





absorbing imidazole derivatives formed by glyoxal have been found to act as a
photosensitizer, initiating radical chemistry under realistic irradiation conditions in the
aerosol phase and initiating aerosol growth in the presence of limonene (Rossignol et
al., 2014).
Shen et al. (2016) revealed that glyoxal can also uptake onto synthetic minerals proxies
of natural mineral dust, forming oligomers, organo-sulphates, formic acid, and glycolic
acid, henceforth suggesting a potential significant mechanism for organic aerosol
formation and modification of the optical and hygroscopic properties of mineral dust.
Following up from the pioneering study by Shen et al. (2016), in this paper we present
the results of laboratory experiments using a large-scale simulation chamber to
investigate the formation of OA from the uptake of glyoxal on realistic airborne mineral
dust particles. Dust aerosols are generated from a natural parent soil from the Gobi
Desert, one of the most important sources of tropospheric dust and representative of
an area where this interaction could take place (Wang et al., 2015).
This paper has two major objectives. First, it provides experimental observations of the
uptake of glyoxal on mineral dust aerosol, leading to the formation of organic aerosol
mass upon interaction and measuring glyoxal uptake coefficient of mineral dust.
Secondly, it presents the chemical composition of the mixed organic-dust aerosols, in
terms of its oxidation state, molecular composition and the evolution of secondary
organic aerosol content from glyoxal.
**2 Experimental**
This study uses the CESAM atmospheric simulation chamber, a 4.2 $m^3$ cylindrical
stainless-steel reactor initially described by Wang et al. (2011). CESAM is specifically
designed to study multiphase processes involving aerosol particles, gas-phase
compounds and water, both in the vapour and liquid phases (Brégonzio-Rozier et al.,
2016; Denjean et al., 2014; Giorio et al., 2017). CESAM is equipped with three 6.5 kW
high-pressure arc xenon lamps (model EX-170GM3-E, IREM SpA, Borgone, Italy) and
6 mm Pyrex plate filters to mimic the solar radiation. A 50 cm stainless-steel four-blade
fan located at the bottom of the chamber ensures a mixing time of about 1 minute for
the gas phase and the homogeneity of the internal composition.
The ageing experiments last up to five hours. Before each experiment, the chamber is
evacuated down to $10^{-4}$ mbar. The chamber is then filled with a mixture of 80% $N_2$



(Messer, purity > 99.995%) and 20% $O_2$ (Linde, 5.0) to an internal pressure exceeding
by about 5 to 10 mbar the local atmospheric pressure, to prevent accidental
contamination during the experiments. For the experiments carried out in wet
conditions, the injection of water vapour precedes the injection of dust. The injection of
glyoxal (1 ppmv) was conducted at least after 30 minutes after the dust to ensure that
the dust particles are homogenously distributed. Irradiation is started within one hour
after the glyoxal uptake onto the particles. Ozone is used to verify the sensitivity of the
reactions to the presence of an oxidant. For those experiments, ozone is injected
before glyoxal.

**2.1 Experimental protocols**

Dust aerosols are generated and injected into the chamber according to the protocol
detailed in Battaglia et al. (2024). The natural soil sample used in this study is from the
Gobi Desert (107.48°N; 36.49°E). Prior use, the soil is sieved at 1000 μm and dried at
100°C for less than an hour to remove adsorbed water and contamination from volatile
gases. A quantity ranging from 30 and 50 g is placed in a 1 L Büchner flask and shaken
at 100 Hz using a sieve shaker (Retsch® AS200) to simulate the saltation and
sandblasting mechanisms through which wind erosion generates airborne dust in the
real atmosphere (Di Biagio et al., 2017). An Aerodynamic Aerosol Classifier (AAC,
Cambustion®) is placed between the dust generator and the chamber to inject mono-
modal dust centred between 300 and 400 nm in geometric diameter.
Glyoxal is prepared by heating a mixture of equal amounts of its trimer hydrate (Fluka®
Analytical) and $P_2O_5$ (Sigma – Aldrich ReagentPlus®, 99%) at 150°C (Horowitz et al.,
2001). The trimer decomposition occurs inside a vial connected to a vacuum gas
manifold. Glyoxal is collected as yellow crystals in a second vial immersed in an
ethanol – liquid nitrogen cold trap at around -90°C and then vaporised in a 2.1 L glass
bulb to a controlled pressure. This vial is connected to the simulation chamber to inject
the glyoxal through a nitrogen flow.
Ozone is generated by a Corona discharge in pure $O_2$ using a commercial dielectric
ozone generator (MBT 802N, Messtechnik GmbH, Stahnsdorf, Germany). Water
vapour is generated by heating ultrapure water (Milli-Q IQ 7000, Merk™) inside a
pressurised stainless-steel vessel, previously rinsed at least three times. The total
organic carbon (TOC) content of the ultrapure water is monitored in each experiment



to evaluate the influence on the production of organic particles, which was found to be
minor (see Text S1 in the Supplementary Material). The relative humidity (RH) inside
the chamber is measured by a HMP234 Vaisala® humidity and temperature
transmitter.

## 2.2 Measurements and instrumentation

### 2.2.1 Gas-phase composition

CESAM is equipped with an in-situ long-path FTIR spectrometer (Bruker Tensor 37),
which enables the collection of spectra with a time resolution of 5 minutes. The spectral
range covered is 700 – 4000 cm$^{-1}$, at a resolution of 0.5 cm$^{-1}$ and an optical path length
of 120 meters. To avoid major water interference, glyoxal is quantified by integrating
the peak corresponding to the stretching of the C–H bonds, in the 2720 to 2930 cm$^{-1}$
interval. Additional species quantified by FTIR spectroscopy are formic acid (HCOOH),
carbon monoxide (CO), and ozone ($O_3$). HCOOH is quantified by integrating the
absorption band centred at 1105 cm$^{-1}$, associated with the C-O bond vibration typical
of the carboxylic group of formic acid. CO is quantified by integrating the typical gas
phase stretching absorption centred at about 2143 cm$^{-1}$, and $O_3$ is quantified by
integrating the absorption band of its asymmetric stretching, centred at about 1043
cm$^{-1}$.
A CAPS (Cavity Attenuated Phase Shift) $NO_2$ analyser (Model T500U, from Teledyne
API), with a concentration range of 10 – 1000 ppbv and an integration time of 30 s was
also connected to the chamber for glyoxal detection. VOCs are monitored by a PTR-
ToF-MS (KORE Technology®, second generation) operated in $H_3O^+$ ionization mode
at a time resolution of 1 minute. The reactor pressure and temperature are 1.35 mbar
and 60°C, respectively, leading to an E/N ratio of 131 Td, where E is the electric field
and N is the concentration of neutral particles. This ratio is used to determine the
effectiveness of the ion-molecule collisions and is given in the unit Townsend (Td). Ion
signals measured by PTR-MS are normalized by signals of reagent ions (i.e. $H_3O+$ and
$H_3O+(H_2O)$) to account for variability in instrumental conditions, following equation 5.2.
Additional gas analysers are used to monitor $NO_x$ (APNA-370 Horiba®; measurement
range 1 – 1000 ppb; sampling flow 0.8 L min$^{-1}$; response time 120 s) and $CO/CO_2$
(APEE ProCeas®; $CO_2$ limit of detection 5 ppm; CO limit of detection 10 ppb; sampling
flow 0.2 L min$^{-1}$; response time 45 s).



**2.2.2 Aerosol total number concentration and size distribution**

The aerosol total number concentration above 2.5 nm is measured by a Condensation Particle Counter Condensation Particle Counter (TSI® model 3075, sampling time 1 s, operated at 1.5 L min$^{-1}$). The aerosol number size distribution is measured by a combination of a Scanning Mobility Particle Sizer (SMPS) consisting of a Differential Mobility Analyser (TSI®, model 3080) coupled with a Condensation Particle Counter (TSI® model 3072, sampling time 180 s, operated at 0.3/3.0 L min$^{-1}$ aerosol flow/sheath flow) measuring particles with mobility diameters between 19.5 and 881.7 nm (107 size channels) and an Optical Particle Counter (sky-GRIMM® OPC model 1.109, sampling flow=1.2 L min$^{-1}$, laser wavelength = 655 nm, sampling time 12 s) measuring particles with optical mobility diameters between 0.265 µm to 31 µm (31 size channels).

The procedure for combining the aerosol size distributions measured by the SMPS and the sky-GRIMM® OPC is based on the method by Baldo et al. (2023), as described in detail in Battaglia et al. (2024). The number size distributions, expressed in d$N$/dlog$D$ (cm$^{-3}$), are used to evaluate the total particle surface S (µm$^2$ cm$^{-3}$) and volume $V$ (µm$^3$ cm$^{-3}$) by assuming spherical particles as

$$S = \int \pi D^2 \frac{\mathrm{d}N}{\mathrm{dlog}D} \ \mathrm{dlog}D \tag{1}$$

$$V = \int \frac{\pi}{6} D^3 \frac{\mathrm{d}N}{\mathrm{dlog}D} \ \mathrm{dlog}D \tag{2}$$

**2.2.3 Aerosol chemical composition**

The aerosol chemical composition is measured by a combination of online and offline methods.

**2.2.3.1 Time-of-flight Aerosol Chemical Speciation Monitor**

A Time-of-flight Aerosol Chemical Speciation Monitor (ToF–ACSM; Aerodyne Research Inc., Billerica, Massachusetts) equipped with a standard vaporiser provides quantitative unitary mass resolution spectra of OA submicronic particles (40 nm–1 µm in vacuum aerodynamic diameter). Particles are sampled with a time resolution of 6 minutes and a flow of 0.85 L min$^{-1}$ through a Nafion membrane dryer (model PD-50T-12) installed upstream of the ToF–ACSM. The incoming aerosol is thermally vaporised



at ~600°C. The resulting gas is ionised by electron impact ionization (EI) and the
fragments are classified by the time-of-flight mass analyser.
Data processing (including mass calibration, peaks integration and air beam correction
of ion intensities) is conducted with Tofware version 3_2_40209, the ACSM data
analysis package for the software Igor Pro 7.08 (Wavemetrics, Inc., Portland, OR,
USA). The organic mass concentration $m_{org}$ is obtained considering a unitary collection
efficiency (CE = 1) and a relative ionization efficiency (REI) of 1.4 (Nault et al., 2023).
This applies also for the organic linked to mineral dust.
Given that the glyoxal fragment $CH_2O^+$ ($m/z$ =30) has an isobaric interference with the
$NO^+$ fragment from nitrate, the contribution of glyoxal to the organic signal at 30 $m/z$ is
estimated with a minor modification of the standard fragmentation table made following
the method proposed by Galloway et al. (2009) in their study of glyoxal uptake on
ammonium sulphate (AS) particles. The contribution to the total signal at 30 $m/z$ from
nitrate is imposed to be 1.7 times the intensity of the nitrate signal at 46 $m/z$, which
corresponds to the 30/46 signal ratio measured during nitrate calibration. The
contribution to 30 $m/z$ of the organic is then the total signal minus the contribution of
the nitrate and the contribution of air. The elemental ratios of the organic fraction O/C
and H/C are calculated from the measured $f_{44}$ and $f_{43}$ respectively, following the
parametrizations proposed by Aiken et al. (2008) and Ng et al. (2011), respectively.
**2.2.3.2. Filter sampling**
Filter samples are collected using a custom-made stainless-steel holder (6 mm
diameter to concentrate particles on a small surface) operated at 10 L min⁻¹ and
preceded by an active charcoal denuder to remove ozone and VOCs. The sampling
time ranges from 30 minutes to 3 hours. Particles are collected on PTFE filters (Zefluor,
47 mm diameter, 2 μm pore size, Pall Life Sciences), and quartz fibre filters
(Tissuquartz 2500 QATUP, 47 mm diameter, Pall Life Sciences). Before sampling, the
PTFE filters and filter holders are cleaned with dichloromethane (99.8 %, HPLC grade)
in an ultrasonic bath. Quartz filters are pyrolyzed at 550°C for approximately 8 hours.
After sampling, filters are folded and placed in an aluminium paper envelope previously
pyrolyzed (same protocol as for filters), and stored in a refrigerator at -18°C. For each
experiment, one blank sample is collected by sampling for about 20 min from the
chamber only filled with $N_2$ and $O_2$. Analytical blanks, corresponding to pyrolyzed filters
that had not undergone any sampling, were also collected.



### 2.2.3.3. SFE/GC-MS organic aerosol analysis

Supercritical fluid extraction coupled with gas chromatography mass spectrometry (SFE/GC-MS) is used to analyse the molecular composition of the aerosol organic fraction. It was originally developed by Chiappini et al., 2006 and was slightly modified by a Teledyne ISCO model 260D pump for the extraction and a GC (Clarus 680 PerkinElmer) –MS (Clarus MS SQ8C Perkin Elmer) for the analysis.

The analytical protocol of the SFE/GC-MS analysis begins by placing the quartz filters inside the extraction cell. Prior to the extraction, 5 µL of two different solutions are deposited on the filters using a precision syringe (CR700-20 1-20ul (22s/2"/3), Hamilton, USA): (a) an internal standard solution composed by 20 µg mL$^{-1}$ of Tridecane (99%, Sigma-Aldrich) and o-Toluic acid (Sigma Aldrich, purity >97 %) in dichloromethane (99.8 %, HPLC grade) and (b) a derivatizing agent solution composed by N,O-bis(trimethylsilyl)trifluoroacetamide (BSTFA) and 1% of trimethylchlorosilane as catalyst, provided by Sigma-Aldrich. The first step of the analysis is a static extraction, in which the cell is filled with supercritical $CO_2$ (LINDE, reference $CO_2$ High Purity)), that interacts with the filter at 300 bar and 60°C, for 40 min. During this step, the trimethylsilylation of hydroxy and carboxy functions by BSTFA also occurs (generating trimethylsilyl (TMS) derivatives). The supercritical fluid containing the analytes is transferred to the GC injector through a deactivated silica transfer line. The injector is cooled at -20°C using liquid nitrogen flowing around the injector for 15 minutes, where the compounds are retained, and the gaseous $CO_2$ is removed. Once the extraction step was completed, the chemical analysis was continued with the injection of the condensed compounds by heat flash on the GC injector from -20°C to 280°C. The compounds are then eluted with helium flowing at 1 mL min$^{-1}$ (Linde) and transferred to the GC (Clarus 680 PerkinElmer) for separation. The temperature gradient of the GC column (Rxi®-5Sil MS column (30 m, 0,25mmi.d., film thickness: 0.25 µm, Restek) goes from 60°C to 280°C at a rate of 5°C min$^{-1}$ and held at 280°C for 10 min. Detection is achieved through electron impact (70 eV electron energy) ionisation followed by a quadrupole mass spectrometer (Clarus MS SQ8C Perkin Elmer) analysis that produces mass spectra from m/z 50 to m/z 300.

The data analysis is conducted using the proprietary software (TurboMass Version 6.1.0.1965 PerkinElmer®). The analysis is limited to the chromatographic peaks which elutes before the internal standards (around 42 minutes), as for higher retention times



the signal to noise ratio is lower and not conclusive. The chromatograms of each filter
sample are compared with those of the analytical and procedural blanks and a mass
spectrum is extracted from each chromatographic peak that is not present in the blank.
To account for the method variability in extraction efficiency concentrations are
corrected using the internal standard o-Toluic acid TMS derivatives. The structural
analysis of the molecule generating every chromatographic peak is then carried out
using two methods. Each mass spectrum is compared with the reference spectra of
the National Institute of Standards and Technology (NIST) Mass Spectral Library
(Version 2.2), which assigns a structure to each spectrum with a relative probability.
For spectra for which the automatic structural assignment fails (low assignment
probability), we searched for target mass fragments derived from molecules linked to
glyoxal reactivity (see Table S1 in the Supplementary Material), in particular the 73 m/z
fragment corresponding to a TMS derivatization $[Si(CH_3)_3]^+$, the 147 m/z fragment
corresponding to two TMS derivatizations $[(CH_3)_2Si=OSi(CH_3)_3]^+$, the 131 m/z fragment
(Glyoxylic acid TMS derivatized – $CH_3$), and 205 m/z fragment (Glyoxal monohydrate
– $CH_3$).

**2.2.3.4. Electrospray ionization (ESI) high-resolution mass spectrometry**

Molecular analysis of the organic fraction collected on the quartz filters is performed
by electrospray ionization (ESI) high-resolution mass spectrometry (Kourtchev et al.,
2015). A high-resolution (mass resolution=100000 at *m/z* 400) LTQ Orbitrap Velos
mass spectrometer (Thermo Fisher, Bremen, Germany) equipped with a TriVersa
Nanomate robotic nanoflow chip-based ESI source (Advion Biosciences, Ithaca NY,
USA) is used to obtain high resolution mass spectra of the methanol extracts following
an adaptation of the procedure described in Kourtchev et al. (2015). Filters are
extracted one time in 1 mL of methanol (Optima TM grade, Fisher Scientific) and two
times in 0.5 mL of methanol under ultrasonic agitation in slurry ice for 15 min. Extracts
are combined and filtered sequentially through a 0.45 µm pore size and a 0.2 µm pore
size Teflon filter (ISODiscTM Supelco), which are then reduced by volume to
approximately 50–200 µL under a gentle stream of nitrogen. The resulting sample is
injected by direct infusion. The negative ionization mass spectra are collected in three
replicates at ranges m/z 50–500 and m/z 150–1000 and processed using Xcalibur 2.1
software (Thermo Scientific). In the settings of the data processing, the following atoms
are included in the peak assignment: C (from 1 to 100 atoms in the possible assigned



molecular formula), H (1-200), O (0-50), N (0-5), S (0-2), $^{13}$C (0-1) and $^{34}$S (0-1). The
allowed mass accuracy in the formula assignment is ± 4 ppm.
The peak assignment to a molecular formula is done according to Zielinski et al. (2018).
The protocol includes internal calibration, noise removal, blank subtraction, and
additional atomic constraints for formula filtering: elemental ratios were set as 0.3
≤H/C≤ 2.5, O/C≤ 2, N/C≤ 0.5, S/C≤ 0.2, $^{13}$C/$^{12}$C≤ 0.011 and $^{34}$S/$^{32}$S≤ 0.045, and
nitrogen rule. In the case of multiple assignments for the same peak, the formula with
the lowest mass error was kept. This process allowed for the retrieval of parameters
describing carbon oxidation, such as O/C and H/C ratios. Consequently, each mass
spectrum was analysed to construct a van Krevelen diagram, which is a graphical
representation illustrating the sample composition in terms of carbon, oxygen, and
hydrogen in the identified molecular formulas (Patriarca et al., 2018). Identified
molecular formulas are categorised into the following groups: CHO, CHON, CHOS,
CHNS, and CHONS.
A targeted search for molecular formulas resulting from glyoxal reactivity is also done.
In this search, we included formulas associated with glyoxal chemical transformations
such as hydration, oxidation, and oligomerization. Starting from the glyoxal formula,
$C_2H_2O_2$, formulas for mono- and dihydration products are $C_2H_4O_3$ and $C_2H_6O_4$,
respectively. Oxidation products included formic acid ($CH_2O_2$), glycolic acid ($C_2H_4O_3$),
glyoxylic acid ($C_2H_2O_3$), and oxalic acid ($C_2H_2O_2$). Oligomers formed by the hydrolysis
of hydrated glyoxal formulas were also sought. In the process of hydrolysis-driven
oligomerization, each successive molecular addition results in the loss of a water
molecule. If the oligomer is a ring, an additional water molecule is lost due to the
condensation of the linear oligomer terminations. Denoting n as the number of
molecules of the monohydrated form ($C_2H_4O_3$) and m as the number of molecules of
the dihydrated glyoxal form ($C_2H_6O_4$) participating in the formation of an oligomer, the
generated linear oligomers will have the formula $C_{2n+2m}H_{4n+6m-2(n+m-1)}O_{3n+4m-(n+m-1)}$,
where the terms -2(n+m-1) and -(n+m-1) in the hydrogen and oxygen atom
stoichiometry indicate water loss from the oligomerization process. For ring oligomers,
formulas characterised by the stoichiometry $C_{2n+2m}H_{4n+6m-2(n+m)}O_{3n+4m-(n+m)}$ are
searched. Similarly, formulas resulting from the condensation of hydrated forms with
the listed organic acids are calculated and researched.
**2.2.3.5. X-ray photoelectron spectrometry (XPS)**



X-ray photoelectron spectrometry (XPS) is used to quantify the elemental O/C ratio of
the particle surface to a depth less than 10 nm. Measurements are performed with a
VG ES-CALAB 250 instrument using monochromatic Al $K_\alpha$ radiation (1486.6 eV). The
O/C ratio is quantified by integrating the areas of $O_{1s}$ and $C_{1s}$ peaks. This last is
contributed by a number of functions, including $-CO_2$, C–O, C–C/C–H as well as C-F
from the Teflon substrate. The contribution of the latter can be evaluated from the F1s
(approximately 690 eV) as described in Denjean et al., (2015). The contribution of $SiO_2$
from the mineralogical composition of the dust to O1s was evaluated by integrating the
Si2p peak (107 eV) and applying the stoichiometric proportions between silicon and
oxygen in the composition of quartz (O/Si = 2). The O/C$_{surf}$ ratio is calculated as follows

$$O/C_{surf} = \frac{n[O_c]}{n[C]} = \frac{(n[O] - 2n[Si])}{n[C]} \qquad (3)$$

Where $n[O_c]$ is the signal coming from oxygen bonded to carbon atoms only, $n[C]$ is
the signal of $C_{1s}$, $n[O]$ the area from $O_{1s}$ and n[Si] is the area of the signal from $Si_{2p}$
that is multiplied by 2 to take into account the silica stoichiometry. The XPS
measurement on a filter collected during one ageing experiment are shown as an
example in Figure S1 in the Supplementary Material.

**2.3 Calculation of the glyoxal uptake coefficient**

The uptake coefficient $\gamma$ is defined as the probability of the gas to be taken up on the
aerosol surface. It is a unit-less parameter expressed by the ratio between the number
of molecules taken up on a surface and the total number of collisions of the gas on the
surface as

$$\gamma = \frac{number\ of\ total\ molecules\ taken\ up}{total\ number\ of\ collisions} \qquad (4)$$

The uptake coefficient $\gamma$ can be calculated in two ways. First of all, it can be estimated
from the first-order heterogeneous loss rate of glyoxal ($k_{het}$, s$^{-1}$) as

$$\gamma = \frac{k_{het}}{\omega} \qquad (5)$$




where ω is the rate of collisions (collision frequency) defined as

$$\omega = \frac{cA_s}{4}$$
(6)

where:
• $c = 146 \times \sqrt{\frac{T}{MW}}$ is the mean molecular speed (m s$^{-1}$), where T is the air temperature
(here 298 K) and MW the molecular weight of the compound of interest (in the case
of glyoxal MW = 58 g mol$^{-1}$).
• $A_s$ is total aerosol surface concentration (m$^2$ m$^{-3}$).
The total aerosol surface concentration ($A_s$) is calculated from the aerosol size
distribution recorded at the end of the dust injection.
The heterogeneous loss rate of glyoxal ($k_{het}$) due to its uptake on dust particles can be
determined as the difference of the loss rate of glyoxal measured during the uptake
experiments ($k_{obs}$) and the glyoxal loss rate on the chamber walls ($k_{loss}$) as

$$k_{het} = k_{obs} - k_{loss}$$
(7)


The glyoxal wall loss is represented by a partition equilibrium described by two first-
order reactions: one for the adsorption of gas phase molecules onto the chamber walls,
and one for the reverse process. The rate constants for both processes have been
obtained experimentally through control experiments with only glyoxal in the chamber
in different relative humidity conditions.
If the uptake reaction is of the first rate, $k_{het}$ is henceforth calculated as
$$k_{het} = \frac{ln\left(\frac{[Gly]_0}{[Gly]_{obs}}\right) - ln\left(\frac{[Gly]_0}{[Gly]_{loss}}\right)}{t}$$
(8)


where [Gly]$_0$ is the initial concentration of glyoxal, [Gly]$_{obs}$ represents the observed
evolution of glyoxal concentration in time, resulting from the sum of uptake and wall





loss, and $[Gly]_{loss}$ represents the estimated glyoxal concentration resulting from the wall
loss.
In addition, $k_{F-OM}$, the rate of formation of the particulate organic matter (POM) on pre-
existing particles following the uptake of glyoxal on the dust can be calculated as

$$k_{F-OM} = \frac{ln\left(\frac{[POM]_t}{[POM]_0}\right)}{t}$$ (9)

where $[POM]_0$ represents the initial POM concentration in the particle phase, and
$[POM]_t$ represents the concentration of the POM formed at a given time.
If the hypothesis that the POM formation is solely due to the uptake of glyoxal, $\gamma$ can
be also evaluated as
$$\gamma = \frac{k_{F-POM}}{\omega}$$ (10)

**3 Results and discussion**
**3.1 Timeline of experiments**
The ageing experiments of monodispersed mineral dust and glyoxal described in this
paper are summarised in Table 1. All the aerosol data are corrected for dilution, wall
loss, and particle loss through the tubing systems as detailed in Text S2 in the
Supplementary Material. Gas phase concentrations are corrected for dilution only.

**Table 1.** Listing and initial conditions of the experiments considered in this study, including experiments
with glyoxal only (experiment type GL), ammonium sulphate and glyoxal (AS + GL), dust only (D) and
dust with glyoxal (D + GL). The glyoxal and ozone gas phase concentrations correspond to the maximum
value measured by FTIR after the respective injections. $V_{seed}$ indicates the maximum volume
concentrations of seed particles (either dust or ammonium sulphate) measured after the particle
injection. The notation "dark/light" indicates experiments when filter samples were collected both in the
dark and with irradiation.

| Experiment type | Reagents | Date | Experiment number | RH, % | Light | $[O_3]$, ppb$_v$ | Temp, K | $[GL]$, ppb$_v$ | $V_{seed}$, µm$^3$ cm$^{-3}$ |
|---|---|---|---|---|---|---|---|---|---|
| Control | GL | 29/04/2021 | $G_1$ | < 5 | dark | --- | 292 | 1130 | --- |
| | | 11/02/2022 | $G_2$ | 77 | light | 1440 | 291 | 627 | --- |
| | AS+GL | 21/02/2023 | $AS_1$ | 38 | dark | --- | 298 | 527 | 50.1 |
| | | 23/02/2023a | $AS_2$ | 35 | dark | --- | 298 | 516 | 48.3 |



| | | 23/02/2023b | $AS_3$ | 32 | light | --- | 298 | 445 | 64.8 |
|---|---|---|---|---|---|---|---|---|---|
| | | 07/09/2023 | $AS_4$ | 81 | light | --- | 301 | 779 | 304.1 |
| | | 08/09/2023 | $AS_5$ | 83 | light | --- | 300 | 430 | 161.2 |
| | D | 31/01/2022 | $D_1$ | < 5 | $^{dark}/_{light}$ | --- | 292 | --- | 31.5 |
| | | 03/02/2022 | $D_2$ | 75% | $^{dark}/_{light}$ | --- | 293 | --- | 55.4 |
| | | 04/02/2022 | $D_3$ | < 5 | $^{dark}/_{light}$ | --- | 293 | 690 | 35.6 |
| | | 08/02/2023 | $D_4$ | 32 | dark | --- | 294 | 940 | 21.5 |
| | | 09/02/2023 | $D_5$ | 31 | light | --- | 295 | 1050 | 52.7 |
| | | 10/02/2023 | $D_6$ | 35 | dark | --- | 294 | 809 | 37.4 |
| | | 13/02/2023 | $D_7$ | 34 | light | --- | 296 | 850 | 51.3 |
| | | 30/04/2021 | $D_8$ | 76 | light | --- | 289 | 759 | 28.3 |
| | | 03/05/2021 | $D_9$ | 79 | light | --- | 290 | 607 | 38.7 |
| Uptake | D+GL | 04/05/2021 | $D_{10}$ | 81 | light | --- | 290 | 371 | 31.5 |
| | | 05/05/2021 | $D_{11}$ | 78 | dark | --- | 291 | 805 | 30.1 |
| | | 06/05/2021 | $D_{12}$ | 82 | dark | --- | 292 | 432 | 21.1 |
| | | 08/02/2022 | $D_{13}$ | 81 | $^{dark}/_{light}$ | 1270 | 293 | 555 | 64.0 |
| | | 09/02/2022 | $D_{14}$ | 78 | $^{dark}/_{light}$ | 1450 | 293 | 756 | 79.8 |
| | | 10/02/2022 | $D_{15}$ | 75 | $^{dark}/_{light}$ | --- | 295 | 600 | 68.4 |
| | | 14/02/2023 | $D_{16}$ | 83 | dark | --- | 296 | 661 | 35.8 |
| | | 15/02/2023 | $D_{17}$ | 75 | light | --- | 298 | 444 | 41.0 |


Table 1 also lists the few control experiments using ammonium sulphate as seed
particles, described in detail in Text S3 in the Supplementary Material. No POM
formation is observed during control experiments with dust or glyoxal only, both dry
and humid conditions and with and without irradiation.
The typical timelines of the particle concentrations (number and volume) and the non-
refractory composition measured in dry conditions and at 30% and 80% relative
humidity are shown in Figure 1.



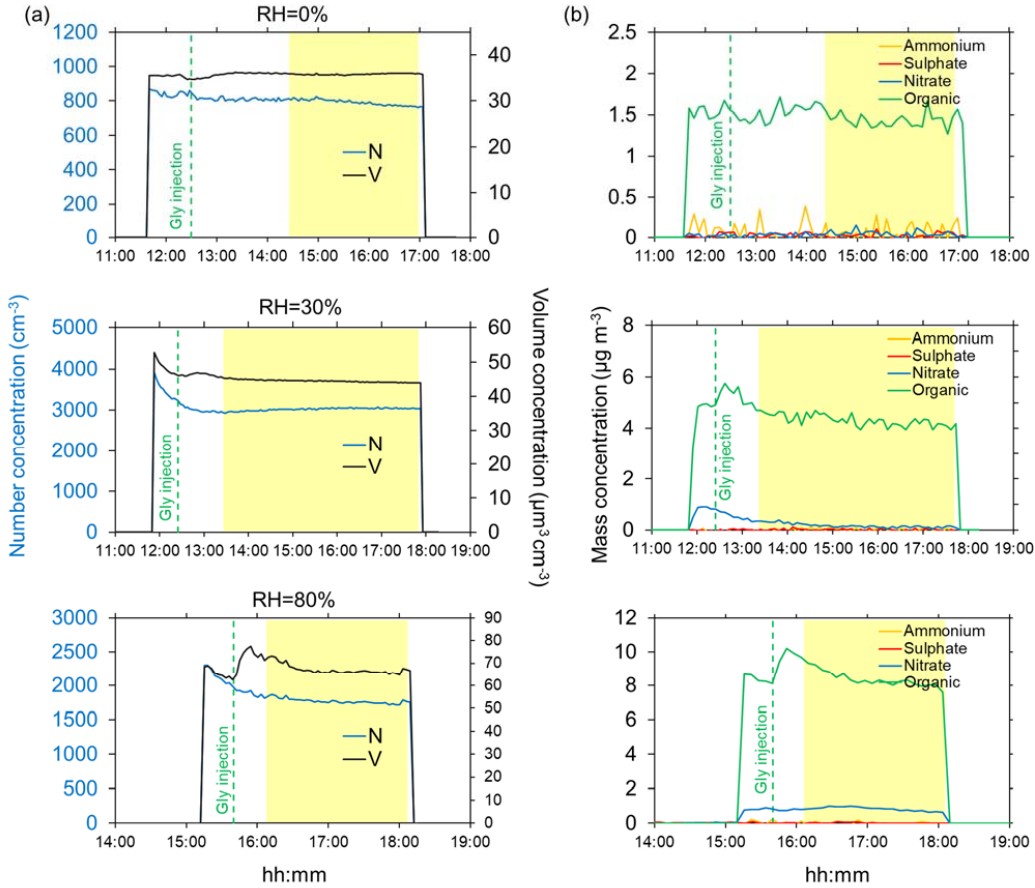

**Figure 1.** Timeline of ageing experiments of submicron dust with gas-phase glyoxal in dry conditions (top, experiment $D_3$), 30% (middle, experiment $D_7$), and 80% RH (bottom, experiment $D_{15}$). Left (a): aerosol total number (N) and volume (V) concentrations (blue and black lines, respectively) calculated from the measured dust size distributions. Right (b): mass concentrations of ammonium, sulphate, nitrate and organic (yellow, red, blue and green lines, respectively) measured by the ACSM. The yellow-highlighted portion of the graph indicates the interval where irradiation takes place, while the green vertical dashed lines indicate the injection glyoxal in the chamber. The dust injection corresponds to the time of the initial increase of the number and volume concentrations. Aerosol time series are corrected for dilution, wall loss and particle loss through the tubings.

Figure 1 shows that in dry conditions, there is no significant variation of either the aerosol number or the volume concentrations, nor the chemical composition (including organics) following the glyoxal injection.

At 30% RH, a small increase in the total volume concentration (approximately 5 µm³ cm⁻³) is observed for about 30 minutes after the injection of glyoxal. This corresponds to an increase of the POM of about 1 µg m⁻³, approximately 20% more with respect to





the value measured before the uptake. On the other hand, the particle number
concentration shows an apparent decrease at the beginning of the experiments,
possibly because the particle loss correction model of Lai and Nazaroff (2000) does
not fully apply to dust particles and humid conditions (see discussion in Battaglia et al.
(2024). After that, and through the duration of the experiment, however, it remains
constant, indicating that the increase in the particle volume occurs on the dust particles
and not because of new particle formation.
At 80% RH, the increase in both the total volume and the POM concentrations is more
pronounced, approximately 10-15 $\mu m^3$ $cm^{-3}$ and 2 $\mu g$ $m^{-3}$, respectively. As for 30% RH,
both total particle volume and the POM concentrations return to values observed prior
the injection of glyoxal, within approximately 30 minutes from their maximum values,
likely due to evaporation. A similar behaviour is observed in the presence of ozone (not
shown). As for 30% RH, the particle number concentration slightly decreases in time
at the beginning of the experiment, but then remains constant, again excluding the
formation of new particles but rather confirming the formation of organic matter on pre-
existing particles. This is also supported by the fact that the rate of increase of POM
and particle volume is the same (slope 3.2 $10^{-4}$ $s^{-1}$ and 3.2 $10^{-4}$ $s^{-1}$, respectively for
POM and total volume), as shown by Figure S2 in the Supplementary Material.
On the other hand, the ratio between the observed increase of the POM (2 $\mu g$ $m^{-3}$) and
that of the particle volume concentration (20 $\mu m^3$ $cm^{-3}$) corresponds to an estimated
mass density of the order of 0.1 $g$ $cm^{-3}$, that is, about 10 times lower than the value of
1 $g$ $cm^{-3}$ expected for glyoxal. This would suggest that part of the organic matter formed
on dust is not detected by the ACSM, as will be further demonstrated in section 3.3.
On the other hand, Figure S2 shows that, after reaching its maximum value, the volume
concentration decreases at a lower rate than the POM (slope 3.9 $10^{-5}$ $s^{-1}$ and 6.1 $10^{-5}$
$s^{-1}$, respectively). This suggests that an additional process could contribute the particle
volume concentration partially compensating the loss of organic matter on the dust
particles.
This is confirmed by Figure 2 comparing the variation in time of the particle volume
distributions, normalised to the total volume, at four different times of the experiments
at 30% and the one at 80%: prior and when the glyoxal is injected, when the POM
reaches its peak value, and at the end of the experiment.



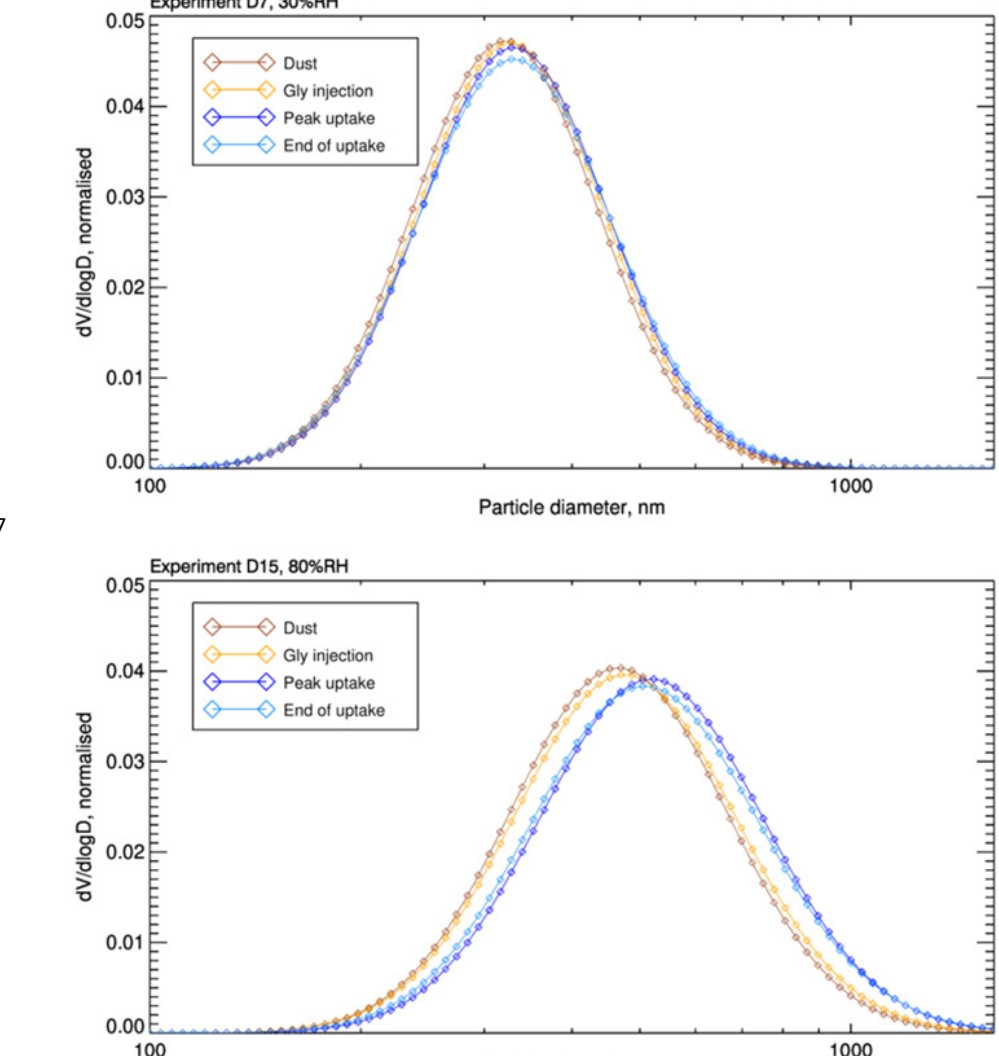

**Figure 2.** Evolution of volume-size distributions for two glyoxal uptake experiments in different relative humidity conditions. The images illustrate the progression of volume-size distributions recorded at four key moments during the experiments. The first distribution (orange) is recorded after the dust is injected into the simulation chamber. The second distribution (yellow) is recorded at the moment of glyoxal injection. The third distribution (blue) corresponds to the peak uptake of glyoxal on the aerosol, and the fourth (light blue) is recorded at the end of glyoxal uptake process. The left image depicts the evolution for the experiment D7 conducted at 30% RH, while the image on the right shows the distributions for the experiment D15 conducted at 80% RH. The results highlight that the distributions grow more significantly at 80% RH, indicating a higher glyoxal uptake and organic formation at elevated humidity levels.

All distributions have a single mode. However, after the injection of glyoxal, the geometric mean volume diameter, measured at the maximum POM concentration,





increases by up to 10% (from 310 to 340 nm) at 30% RH, and up to 20% (from 450 to
540 nm) at 80% RH. Interestingly, even at the end of the experiment, when the POM
concentration returns to its initial value, the increase in geometric mean diameter of
the aerosol is irreversible. This effect could be explained by the hypothesis that the
uptake of glyoxal enhances the dust hygroscopicity. After glyoxal uptake, the particle
becomes more hygroscopic and the difference in total volume between the beginning
and end of the experiment is due to water uptake which adds up to the formed organic
aerosol mass.
Finally, Figure 1 shows also that, while sulphate and ammonium are never detected, a
background concentration of nitrate up to 1 μg m$^{-3}$ is measured by the ACSM as soon
as the dust particles are injected in the presence of water. We attribute it to the
heterogeneous interaction between $NO_2$ and the dust particles (Goodman et al., 1999)
as indeed, a background concentration of a few ppb of $NO_2$ is present in the chamber
as a result of the procedure used to reduce the TOC content in the injected water (see
Figure S3 in the Supplementary Material). However, since the contribution of nitrate
represents at maximum 1% of the injected dust mass and whether decreases or
remains constant throughout the experiment, its contribution to the particle growth and
overall ageing of the mineral dust should be negligible.
Figure 3 shows the time series of the gas-phase compounds detected during the same
experiment (D$_{15}$) at 80% RH.





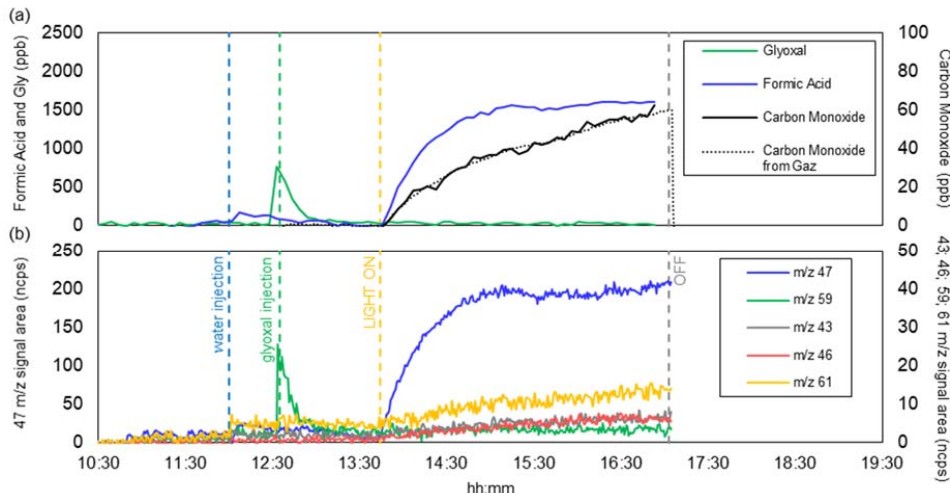


**Figure 3**. Time series of the gas-phase composition observed during experiment $D_{15}$: (a) concentrations
of carbon monoxide, glyoxal and formic acid measured by FTIR (for CO the measured of the online
analyser are also shown); (b) various VOC ions (m/z 47, 59, 43, 46 and 61) measured by the PTR-MS.
Ion signals measured by PTR-MS are normalized by signals of reagent ions (i.e. $H_3O^+$ and $H_3O^+(H_2O)$)
and therefore expressed in normalized counts (ncps). The blue vertical dashed lines indicate the
injection of water in the chamber; the green vertical dashed lines indicate the injection of glyoxal in the
chamber, the yellow dashed lines indicate the beginning of irradiation and the grey dashed lines indicate
the end of irradiation.

550

The measured glyoxal concentration after the injection (Figure 3a) is lower than the
nominal concentration of 1 ppm and goes to zero within minutes due to the rapid
interactions with the walls of the chamber, water vapour, and the dust particles. Upon
irradiation, formic acid and carbon monoxide are formed, as expected by the photolysis
of glyoxal (De Haan et al., 2020). Fragments m/z = 46 and 47 are observed during
water injection and photolysis, and could originate from the deprotonated and
protonated form of formic acid, respectively. This suggests that a minor fraction of the
formic acid could result from the desorption of compounds (including glyoxal) from the
chamber walls. Fragments m/z = 43 and m/z = 61, and occasionally m/z = 45 (not seen
during experiment $D_{15}$ and therefore not shown in Figure 3b), are observed at a
normalised intensity two orders of magnitude lower than that of formic acid, but not
attributed. The quantification with both PTR-MS and FTIR in our experimental RH
conditions is complicated by the presence of water, which reduces the sensitivity of
PTR-MS and can interfere with the absorption of various organic compounds, making
their quantification less accurate.



## 3.2 Glyoxal uptake coefficient

For experiment $D_{15}$ at 80% RH, Figure 4 shows the temporal evolution of the natural logarithm of the glyoxal concentration measured by the FTIR, compared to that measured during a typical blank experiment without dust particles. Figure 4 additionally shows the variation of the aerosol organic fraction measured by the ACSM during the same time period.

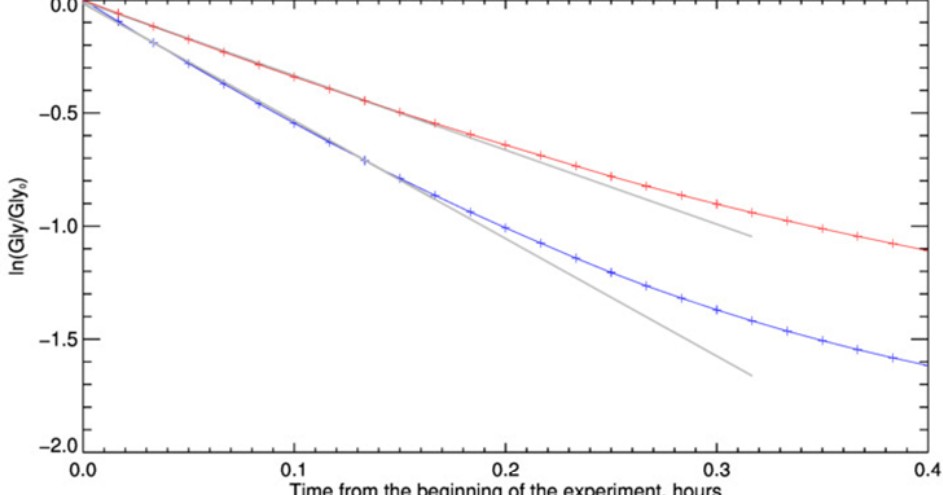

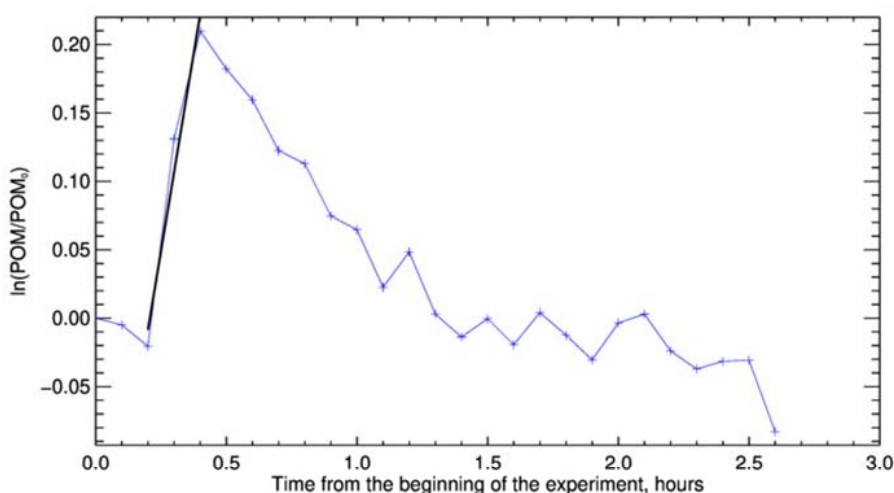

**Figure 4**. Measurement of glyoxal uptake coefficient on dust for the experiment $D_{15}$, conducted at 80% RH. The figure compares the two methods for measuring the uptake coefficient. In the top image, results are shown for the method based on monitoring the decay of gas-phase glyoxal. The red and blue curves



represent the logarithm of the ratio between the calculated decay of gas-phase glyoxal in the absence
and presence of dust aerosols, respectively. The black lines represent the linear fit whose slope provides
the heterogeneous kinetic constants of the two processes. The image at the bottom displays the result
of the uptake coefficient measurement for the same experiment, obtained from the organic formation on
the aerosol monitored by ACSM. The blue time series shows the logarithm of the ratio between the
measured organic concentration divided by the initial organic on dust aerosol, while the black line is the
linear fit representing the kinetics of organic formation.

Within the first 10 minutes after the injection of glyoxal, the decrease of the natural
logarithm concentrations ratio with time in the presence of dust is linear (the rate is
constant). After that, the loss slightly deviates from linearity. The difference from
linearity is more evident for the blank experiment, when it occurs earlier than when the
dust is present. These observations indicate that, within the first 10 minutes, the uptake
of glyoxal on the dust particles can be considered to follow a first-order kinetic and its
rate represents an initial uptake coefficient. In the following 20 minutes approximately,
the uptake slows down, possibly because all the sites available on the particle surface
become occupied, but also because that desorption from the particle surface could
reinject glyoxal in the reactive mixture. On the particle phase, the natural logarithm of
the organic concentration, normalised by its initial value increases rapidly and linearly,
almost on the same time scale of that of the loss of glyoxal, but then decreases to
return to its initial value within approximately one hour. These observations confirm
that the uptake of glyoxal results in a formation of OA on the dust particles, but that
this process is reversible.
The uptake coefficients calculated as the linear fit of the glyoxal and particle organic
concentration are presented in Table 2.

**Table 2.** Uptake coefficients for glyoxal on mineral dust and ammonium sulphate calculated from the
loss of gas-phase glyoxal and the rate of OA formation for the experiments conducted at 80% RH. The
initial glyoxal concentration is reported. The aerosol surface concentration ($A_s$) corresponds to the value
preceding the glyoxal injection. Ozone concentration is the maximum concentration measured by FTIR
spectroscopy after the injection. For ammonium sulphate, only the γ values calculated from the loss of
gas phase glyoxal are presented, as the ACSM collection efficiency (CE) for ammonium sulphate varies
significantly during OA formation (Matthew et al., 2008).

| Date | Experiment ID | RH% | $[GL]_0$, $ppb_v$ | Ozone (ppb) | $A_S$ ($m^2$ $m^{-3}$) | $\omega$ ($s^{-1}$) | $\gamma_{gas}$ | $\gamma_{OA\ glyxoal}$ |
|---|---|---|---|---|---|---|---|---|
| 30/04/2021 | $D_8$ | 76 | 759 | --- | $4.8\ 10^{-4}$ | $3.9\ 10^{-2}$ | $6.0\ 10^{-3}$ | $1.0\ 10^{-3}$ |
| 03/05/2021 | $D_9$ | 79 | 607 | --- | $6.1\ 10^{-4}$ | $5.0\ 10^{-2}$ | $1.5\ 10^{-2}$ | $1.5\ 10^{-2}$ |
| 04/05/2021 | $D_{10}$ | 81 | 371 | --- | $5.1\ 10^{-4}$ | $4.2\ 10^{-2}$ | $1.7\ 10^{-2}$ | $9.0\ 10^{-3}$ |
| 05/05/2021 | $D_{11}$ | 78 | 805 | --- | $4.6\ 10^{-4}$ | $3.8\ 10^{-2}$ | $8.0\ 10^{-3}$ | $5.0\ 10^{-3}$ |
| 06/05/2021 | $D_{12}$ | 82 | 432 | --- | $3.5\ 10^{-4}$ | $2.9\ 10^{-2}$ | $1.2\ 10^{-2}$ | $2.3\ 10^{-2}$ |
| 08/02/2022 | $D_{13}$ | 81 | 555 | 1270 | $7.1\ 10^{-4}$ | $5.8\ 10^{-2}$ | $4.0\ 10^{-3}$ | $4.0\ 10^{-3}$ |
| 09/02/2022 | $D_{14}$ | 78 | 756 | 1450 | $8.5\ 10^{-4}$ | $7.0\ 10^{-2}$ | $4.0\ 10^{-3}$ | $5.0\ 10^{-3}$ |
| 10/02/2022 | $D_{15}$ | 75 | 600 | --- | $8.4\ 10^{-4}$ | $6.9\ 10^{-2}$ | $1.0\ 10^{-2}$ | $5.0\ 10^{-3}$ |





| 14/02/2023 | $D_{16}$ | 83 | 661 | --- | $6.0\ 10^{-4}$ | $4.9\ 10^{-2}$ | $4.0\ 10^{-3}$ | $1.5\ 10^{-2}$ |
|---|---|---|---|---|---|---|---|---|
| 07/09/2023 | $AS_4$ | 81 | 779 | --- | $6.3\ 10^{-3}$ | $5.2\ 10^{-1}$ | $9.8\ 10^{-4}$ | --- |
| 08/09/2023 | $AS_5$ | 83 | 430 | --- | $2.0\ 10^{-3}$ | $1.7\ 10^{-1}$ | $1.2\ 10^{-3}$ | --- |
| Average dust | | | | | | | $9\ (\pm 5)\ 10^{-3}$ | $9\ (\pm 7)\ 10^{-3}$ |
| Average AS | | | | | | | $1.1\ (\pm 0.2)\ 10^{-4}$ | |


The average uptake coefficients for glyoxal on the Gobi mineral dust calculated at 80%
RH from the gas-phase uptake and the particle formation are $\gamma_{Gly-Dust-gas} = 9 \times 10^{-3}$
(standard deviation ± 5) and $\gamma_{Gly-Dust-OA} = 9 \times 10^{-3}$ (standard deviation ± 7), respectively.
The two average values agree. This suggests that every glyoxal molecule in the gas
phase is taken up by the airborne dust particles. This suggests also that the uptake
occurs on airborne particles only, as expected as the dust particles are selected in the
submicron range and that minimal deposition of dust particles is observed in the first
30 minutes after injection. The primary mechanism of particle loss during this period is
dilution, which does not interfere with uptake. The standard deviations of the mean
values are large, being attributed to the fact that the state of the chamber walls and the
dust size distribution vary from one experiment to the other, and that the
aerosol/chamber walls surface ratio is very low ($0.08\text{-}1.5\ 10^{-3}$). The presence of ozone
appears uninfluential.
The results of the current study can be compared with the literature. Shen et al. (2016)
investigated the uptake of glyoxal on mineral proxies, i.e. $SO_2$ and $CaCO_3$ under
various levels of relative humidity. These authors determined the uptake coefficients
after a long exposition of the surface to glyoxal (steady state uptakes) and found that
the uptake coefficients are reduced with increasing the gas phase concentration of
glyoxal. At 1 ppb concentration and a relative humidity of 60% the uptake coefficients
determined on suspended particles of calcite ($CaCO_3$, $\gamma = (1.4 \pm 0.1) \times 10^{-4}$) and
alumina ($AlO_3$, $\gamma = (5.5 \pm 0.1) \times 10^{-5}$). Our values are measured in a shorter time frame
and correspond to an initial uptake of glyoxal. They are approximately one order of
magnitude higher than those obtained by Shen et al. (2016). These authors scaled the
uptake coefficient to the specific surface area of the dust, which henceforth could
correspond to a lower limit. On the contrary, in our case, we use a geometric surface
area (assuming spherical particles) which could lead to an overestimation of the uptake
coefficient.



Zogka et al. (2024) used a Knudsen cell to evaluate the initial and steady state glyoxal
uptake coefficient bulk soil samples of various origins. At low relative humidity, these
authors found that for Gobi soil sieved to less than 63 μm in diameter the initial uptake
coefficient using the geometric surface area was 0.18 (corresponding to an upper limit
of the uptake) independent of glyoxal concentration. However, the steady state uptake
coefficients determined after a long processing of surface were found to decrease with
increasing glyoxal concentration, due to aging of the surface.
Various reasons can explain these apparently different results. First of all, in CESAM
we measure the initial uptake coefficient at humid conditions, which is independent of
concentration. On the other hand, Shen et al. (2016) measured steady-state uptake
coefficient at lower glyoxal concentrations (< 1 ppb) than we did (>400 ppb). As shown
by both Shen et al. (2016) and Zogka et al. (2024), the steady-state uptake coefficient
decreases with the concentration of glyoxal, regardless of the relative humidity.
Secondly, the uptake coefficient is inversely proportional to the available particle
surface, which in our case is smaller than Zogka et al. (2024) who used soils sieved to
63 μm. Shen et al (2016) used standard mineral particles of various sizes from 35 nm
to 5 μm, while our particle size distribution peaked between 300 and 400 nm. Shen et
al. (2016) used single minerals, while Zogka et al. (2024) and our study share the same
soil sample from Gobi. While the uptake coefficient should depend on the dust
mineralogy, this is difficult to ascertain in the present study.
Overall, although the experiments performed in literature with those of the current study
were performed under different conditions, results indicate that natural Gobi dust is an
effective sink of glyoxal, with initial uptake coefficient independent of glyoxal
concentration, pointing to a first order removal process. However, the long-term ageing
of particles leads to lower uptake coefficients that strongly depend on glyoxal
concentration due to the depletion of surface sites.
**3.3. Organic composition**
This section discusses the chemical composition of the organic matter on mineral dust
following the interaction with glyoxal. Details on the organic composition of the native
dust are provided in Text S4 in the supplementary material.





### 3.3.1. Measurements by the ACSM

Figure 5 shows the time evolution of the intensity of the organic fragments detected by the ACSM at 80% RH (experiment $D_{15}$) before the injection of glyoxal (dust only), during the POM formation, after the maximum concentration, and at the end of the experiment, when it returned to its initial value.



674

**Figure 5.** ACSM organic mass spectra (intensities normalized to the total organic concentration) recorded during the experiment $D_{15}$: (a) before glyoxal uptake (dust organic fraction composition), (b) during glyoxal uptake, (c) after reaching the maximum uptake on the particles and (d) 1h later under irradiation. Panels on the left show the mass spectra ranging from 10 to 60 m/z, while the panels on the right represent fragments from 60 to 200 m/z (their intensity is approximately one hundred times lower). The inserts display the time series of organic concentrations measured by the ACSM. A black arrow indicates the time corresponding to the mass spectrum shown. The yellow-highlighted shaded area indicates the interval where irradiation takes place, while the green vertical dashed lines indicate the moment of glyoxal injection in the chamber.






Fragments at 28 m/z and 44 m/z, typical of oxidised compounds, are ubiquitous at all
stages of the experiment. Their relative intensity follows the kinetic of the uptake,
decreasing during the formation of organic aerosols and reverting to their initial value
towards the end of the experiments. Fragment 69 m/z, attributed to $C_3HO_2^+$ and
$C_4H_5O^+$, has the highest intensity among ions above m/z 60 in all four spectra,
suggesting that it is not related to glyoxal reactivity. Galloway et al. (2009) observed
this fragment and identified it as a nitrogen-containing organic molecule with a formula
$C_3H_3NO^+$ from the reaction of glyoxal with ammonia during the uptake of glyoxal on
ammonium sulphate. This fragment was also observed at significant intensities in
organic aerosols originating from phenolic derivatives, such as guaiacol and syringic
acid, metabolites of plants (Sun et al., 2010), and could be explained as soil residues
from vegetation but also animals (Nieder et al., 2018).
The signal of fragments at 29 m/z ($CHO^+$), 30 m/z ($CH_2O^+$), and 58 m/z (molecular
peak $C_2H_2O_2^+$), characteristics of glyoxal, appear during the uptake (second panel from
the top in Figure 5) but their relative intensity decreases with time. Fragments m/z =
105 and m/z = 131 observed during both the uptake and the photolysis are specific
markers of the interaction between dust and glyoxal (Liggio et al., 2005). The former is
attributed to an oligomeric structure of glyoxal generated by the condensation of two
molecules of glyoxal hydrate (see also Table 3 in Liggio et al. (2005)). The latter is not
identified, yet it consistently accompanies the glyoxal's primary ions. In photo-oxidation
experiments of glyoxal in the aqueous phase, Carlton et al. (2007) found that its
abundance increased proportionally with the increase in glyoxal concentration. In our
study, this fragment is detected in all conditions (with and without light or ozone).
Fragments at 119 and 120 m/z have been observed in organic aerosols derived from
isoprene and attributed to an organic acid with the formula $C_8H_8O_4$ (m/z 120) and its
deprotonated form (Safi Shalamzari et al., 2013). In our experiments, they consistently
increase after glyoxal uptake, and in particular upon irradiation. Although it is not
straightforward to assign these fragments to a unique glyoxal derived formula, we
hypothesise that they may arise from oxidised forms of oligomers.
Upon irradiation, m/z = 147 is accompanied by m/z = 165. These fragments have been
observed in organic aerosols produced in experiments of aqueous phase glyoxal
oxidation via OH radicals (Lim et al., 2010). The fragment m/z 165 is attributed to the





condensation of one molecule of glyoxal di-hydrate with one molecule of oxalic acid or
two molecules of glyoxylic acid hydrate. The fragment m/z 147 could result from the
dehydration of the aforementioned products. These fragments are present with similar
intensity in the spectrum of native dust, and their increase in intensity under irradiated
conditions might therefore be due to the reversibility of the interaction rather than an
oxidation process of glyoxal.
The occurrence of the fragment at 18 m/z is often resulting from the loss of a water
molecule ($H_2O$) from hydrated organic compounds. The slight decrease of the intensity
of this fragment during glyoxal uptake could therefore be explained by the presence of
the oligomerization of hydrated glyoxal molecules. This process leads to the loss of
two hydroxide groups for each added molecule in favour of the formation of acetal or
hemiacetal bonds in the structure of the resulting newly formed secondary organic
aerosol.
**3.3.2. Molecular analysis**
The list and conditions of the samples analysed by SFE/GC-MS and ESI-Orbitrap are
reported in Table S2 in the Supplementary Material. The summary of the organic
molecules detected by those analysis is presented in Table 3 and discussed in the next
paragraphs.

**Table 3.** *Summary of frequently observed compounds identified by SFE/GC-MS analysis and glyoxal-*
*related formulas observed with ESI-Orbitrap, along with the suggested structures under the different*
*experimental conditions tested.*

| Molecular formula | Name | Tentative Structure | Technique | Experimental conditions |
|---|---|---|---|---|
| $C_2H_2O_3$ | Glyoxylic acid | | ESI-Orbitrap | Dust+Gly, 80%, Dark, $O_3$ |
| $C_2H_4O_3$ | Glycolic acid | | ESI-Orbitrap SFE/GC-MS | Dust+Gly, 30%, Dark<br>Dust+Gly, 30%, Light<br>Dust+Gly, 80%, Light<br>Dust+Gly, 80%, Light, $O_3$ |
| $C_2H_2O_4$ | Oxalic acid | | ESI-Orbitrap | Dust+Gly, 80%, Light |
| $C_2H_4O_4$ | Glyoxylic acid monohydrate | | ESI-Orbitrap SFE/GC-MS | Dust+Gly, 80%, Light |
| $C_3H_6O_3$ | Lactic acid | | SFE/GC-MS | Dust, 80%, Light<br>Dust+Gly, Dry, Dark<br>Dust+Gly, 30%, Dark<br>Dust+Gly, 30%, Light<br>Dust+Gly, 80%, Light<br>Dust+Gly, 80%, Dark, $O_3$<br>Dust+Gly, 80%, Light, $O_3$ |



| | | | | |
|---|---|---|---|---|
| $C_5H_8O_3$ | Levulinic acid | | SFE/GC-MS | Dust, Dry, Dark<br>Dust, 80%, Dark<br>Dust, Dry, Light<br>Dust+Gly, 30%, Dark<br>Dust+Gly, 30%, Light<br>Dust+Gly, 80%, Dark, $O_3$<br>Dust+Gly, 80%, Light, $O_3$ |
| $C_7H_8O$ | Benzyl alcohol | | SFE/GC-MS | Dust, 80%, Dark<br>Dust+Gly, 30%, Dark<br>Dust+Gly, Dry, Light<br>Dust+Gly, 30%, Light<br>Dust+Gly, 80%, Light |
| $C_9H_{16}O$ | Cyclohexanone, 3,3,5-trimethyl | | SFE/GC-MS | Dust, 80%, Dark<br>Dust+Gly, 30%, Dark<br>Dust+Gly, 30%, Light<br>Dust+Gly, 80%, Light, $O_3$ |
| $C_4H_4O_4$ | Glycolic acid dimer | | ESI-Orbitrap | Dust+Gly, 80%, Light |
| $C_4H_6O_5$ | Glyoxal oligomer | | ESI-Orbitrap | Dust+Gly, 80%, Light |
| $C_6H_6O_6$ | Glyoxal oligomer | | ESI-Orbitrap | Dust+Gly, 80%, Light |
| $C_8H_{16}O_{12}$ | Glyoxal oligomer | | ESI-Orbitrap | Dust+Gly, 80%, Light |
| $C_{10}H_{12}O_{11}$ | Glyoxal oligomer | | ESI-Orbitrap | Dust+Gly, 80%, Dark |


### 3.3.2.1. SFE/GC-MS analysis

Figure 6 shows the comparison of four gas chromatograms obtained for experiments D$_2$, D$_{13}$, D$_{15}$ and D$_{16}$ at 80% RH, with and without irradiation, with and without ozone (Table 1).

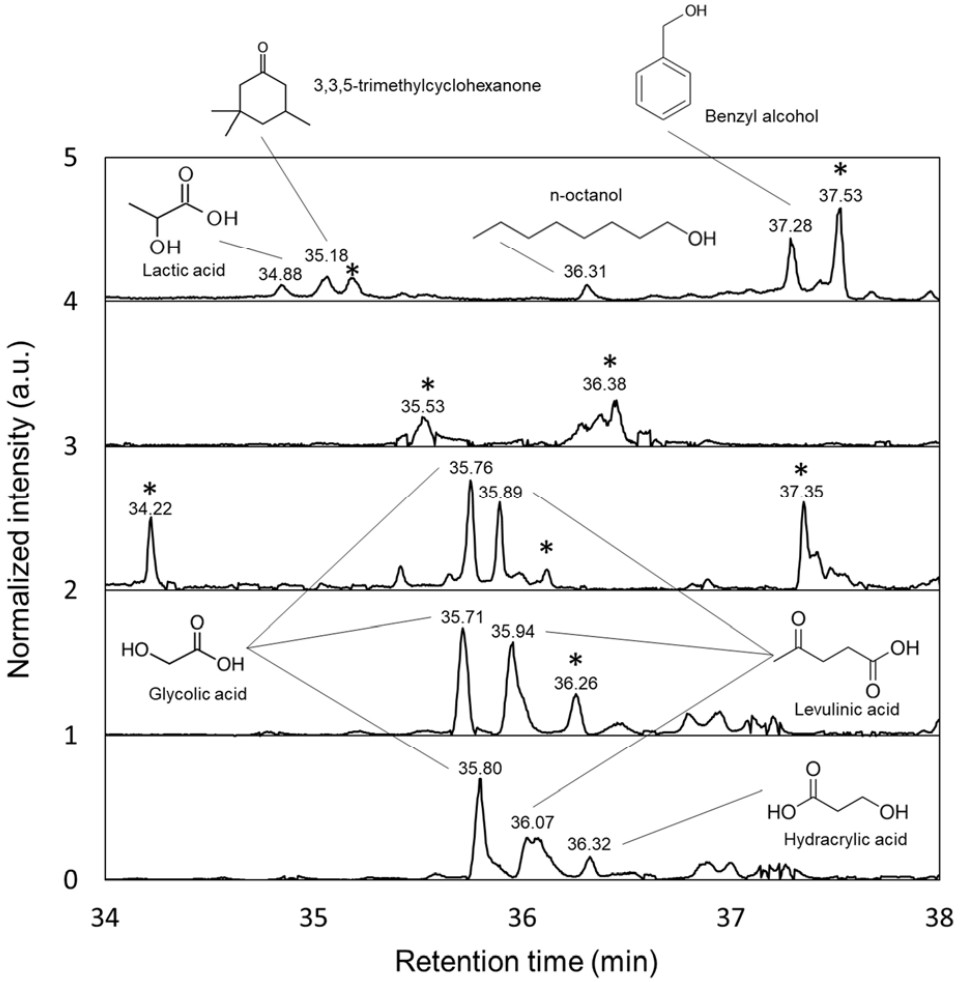

744

**Figure 6.** SFE/GC-MS chromatograms recorded from filters collected during one dust control experiment and four glyoxal uptake experiments under four different conditions. From up to bottom: the first chromatogram is from Experiment $D_2$, dust control experiment under 80% RH and irradiation. The second is from Experiment $D_{16}$ with dust and glyoxal at 80% RH under dark conditions. The third is from Experiment $D_{15}$ with dust and glyoxal at 80% RH under irradiated conditions. The fourth and fifth chromatograms are from Experiment $D_{13}$ with dust and glyoxal at 80% RH in the presence of ozone under dark and irradiated conditions, respectively. The two peaks of higher intensity appearing after 41 minutes are from the internal standards added to the solution: tridecane around 41 minutes and ortho-toluic acid at about 41.8 minutes. The intensity is normalized to peak of the internal standard ortho-toluic acid. The chromatograms start at 32 minutes as 15 minutes are required for the removal of $CO_2$ from the extraction fluid, and approximately another 15 minutes represent the delay required for the solvent to pass through the column and reach the electron ionization (EI) MS detection. For the second and third spectra, is noticeable that under irradiated conditions the number of peaks increases significantly compared to dark conditions, likely due to enhanced chemical reactions driven by light. This effect appears to be less pronounced in the presence of ozone.

760





The chromatograms of samples collected in the presence of glyoxal generally exhibit a higher number of peaks compared to samples of dust-only, indicating glyoxal oxidation and production of organic aerosols. A higher number of peaks are detected in samples after irradiation compared to dark conditions (see Figure 6). This suggests that the exposure to light influences the chemical composition of the samples by promoting pathways that alter the chromatogram profile, possibly through photochemical reactions, which could lead to glyoxal oxidation products. However, in the presence of ozone and glyoxal, the chromatogram profile recorded under dark conditions is comparable to that recorded in the presence of light, suggesting that ozone may play a significant role in the oxidation process, driving similar chemical reactions in both light and dark environments. This could imply that the oxidative capacity of ozone is sufficient to promote glyoxal oxidation and organic formation independently of photolytic processes, resulting in comparable chromatogram profiles regardless of the presence of light.

The compounds identified using SFE/GC-MS primarily consist of carboxylic acids and relatively light-weight carbonyl compounds (<150 Da). Lactic and levulinic acids are detected in 10 and 9 samples, respectively, regardless of the experimental conditions. Compounds detected less frequently include benzyl alcohol, 3,3,5-trimethylcyclohexanone, methylphosphonic acid (5 samples), decanal (4 samples), and various organic acids, including heptanoic, propanedioic, and hydroacrylic acid (3 samples), as well as benzoic acid and 1-octanol (2 samples).

Glycolic acid and is only detected during experiments with glyoxal. Light and the presence of ozone seem to favour its formation. Indeed, the fourth panel in Figure 6 suggests that ozone might substitute light in promoting the formation of glycolic acid from glyoxal at high RH, suggesting an alternative oxidative pathway. Glycolic acid is also detected at 30% RH (not showed), both with and without light, in agreement with the experiments on dust by Shen et al. (2016), but differently than reported by Galloway et al. (2009) on ammonium sulphate. Monohydrated glyoxylic acid is found in one sample at 80% RH under irradiated conditions, likely due to the known pattern of oxidation of glyoxal and glycolic acid with OH radicals (Buxton et al., 1997).

Twelve mass spectra have a profile unrecognised by the NIST library. An example is shown in Figure 7.



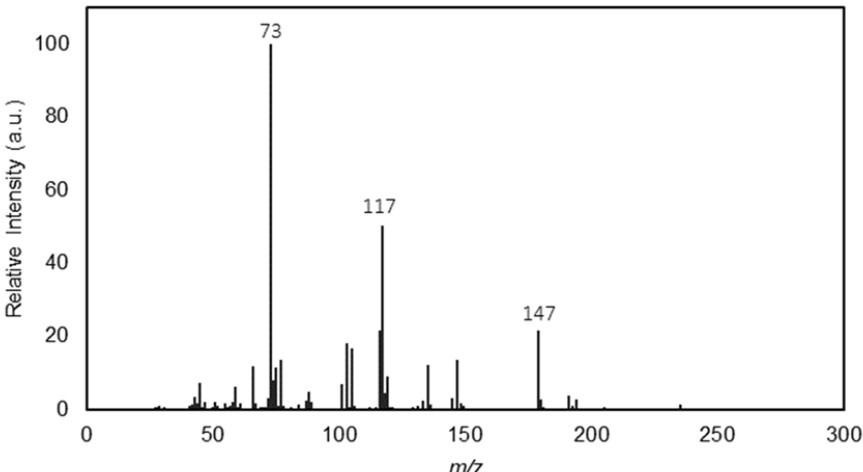

**Figure 7**. SFE Electron ionization mass spectrum recorded from the filter collected during the experiment $D_{13}$ (80% relative humidity in the presence of $O_3$), where the characteristic peaks of the TMS functionalization of two hydroxyl groups (74 and 147 m/z) and one carboxylic group (117 m/z) are observed. The retention time of the peak corresponding to this mass spectrum was 39.58 min.

Notably, ten are collected under humid conditions (both 30% and 80% RH) and in the presence of light. Eleven are characterised by 73 m/z $[Si(CH_3)_3]^+$ for at least one functionalization, 147 m/z $[(CH_3)_2Si=OSi(CH_3)_3]^+$ for at least two, and their multiples for a greater number of hydroxyl functionalities (-OH). In addition, the fragment at 117 m/z $[COO=Si(CH_3)_3]^+$ is detected. These compounds are attributed to trimethylsilyl-multi-functionalised molecules from small oligomers of hydrated glyoxal (multiple functionalities) that have undergone partial oxidation, as indicated by the presence of carboxylic group peak in most of the spectra (117 m/z).

**3.3.2.2. ESI-Orbitrap**

On the 15 samples analysed by ESI-Orbitrap (Table S2 in the Supplementary Material), signals attributable to products of the oxidation, hydration, or oligomerization of glyoxal are found only for experiments at high relative humidity (80%), both in the dark but mostly in irradiated conditions. Figure 8 illustrates the mass spectrum and the assigned formula for experiment $D_{10}$.



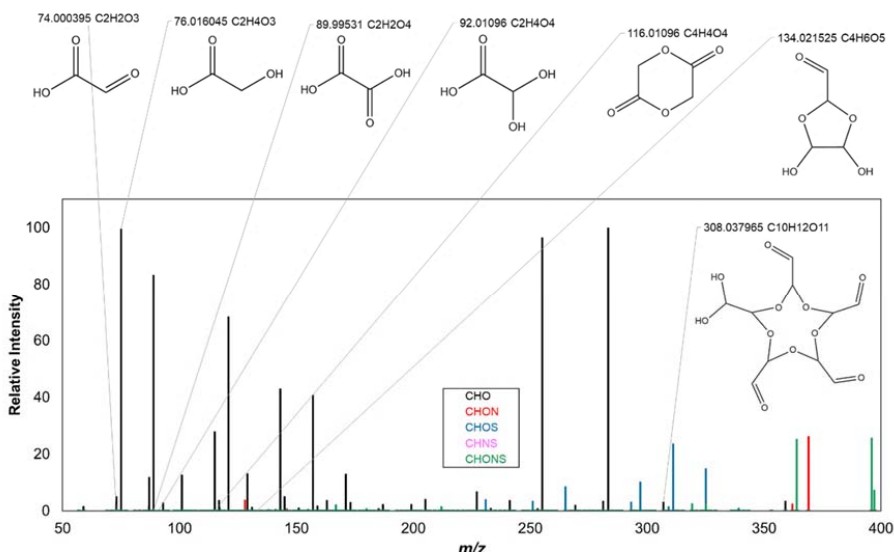

**Figure 8**. ESI Orbitrap MS from the filter $D_{10}$: uptake of glyoxal on mineral dust at 80% RH and under irradiation. In the mass spectra, the peaks referring to formulas for which is possible to suggest a structure from glyoxal reactivity are labelled.

Molecules corresponding to oxidised compounds are predominantly observed on filters collected under irradiated conditions: $C_2H_4O_3$ (also detected by SFE/GC-MS) attributed to glycolic acid; $C_2H_2O_4$, attributed to oxalic acid, and $C_4H_4O_4$, attributed to a dimer of glycolic acid. Its monohydrated form, $C_2H_4O_4$ (also detected by SFE/GC-MS) is also observed in dark conditions. In the presence of ozone, glyoxylic acid is also observed in dark conditions. Particle phase formic acid, observed on both dust and ammonium sulphate by various authors (Galloway et al., 2009; Rubasinghege et al., 2013; Shen et al., 2016), is not detected in our study neither by molecular analysis nor the ACSM. Although the reasons remain unclear, Shen et al. (2016) suggested that, above approximately 50% RH, adsorbed water could compete for surface reactive sites resulting in suppressing the formation of organic acids onto the dust particles. In contrast, the formation of glycolic and glyoxylic acid appears to be less affected by the presence of adsorbed water, as they are detected solely in humid conditions. This is possibly due to differences in their chemical pathways or their interactions with the dust surface. These acids may form through mechanisms that are less competitive with water adsorption, or their precursors have a higher affinity for the reactive sites on dust



particles (possibly due to the presence of two carbonyl regions) allowing their formation
to proceed even in the presence of high humidity.
Compounds from $C_4$ to $C_{10}$, oligomerization products of the glyoxal mono and di
hydrate forms, are observed only at 80% RH. The following peaks are detected and
attributed: $C_4H_6O_6$ (1 monohydrated glyoxal + 1 dehydrated glyoxal forming a 5-atom
ring), $C_8H_{16}O_{12}$ (4 dehydrated glyoxal molecules forming an 8-membered ring), and
$C_{10}H_{12}O_{11}$ (4 monohydrated glyoxal molecules + 1 dehydrated glyoxal molecule
forming a ring structure). The oligomer $C_6H_6O_6$ (3 molecules of monohydrated glyoxal
forming a 6-membered ring) is detected only under dark conditions. $C_8H_{16}O_6$
corresponds to an oligomer previously observed by Shen et al. (2016) on mineral dust.
Based on the observations above, Figure 9 shows the suggested chemical
mechanism.

**Figure 9.** Proposed reaction scheme to explain the glyoxal-related molecular formulas detected through
ESI-Orbitrap mass spectrometry and SFE/GC-MS.





The primary oligomers detected are attributed to the condensation of hydrated glyoxal forms specifically the mono- and di-hydrated glyoxal forms, through hydrolysis. The oligomer V is attributed to the condensation of two glycolic acid molecules.

**3.4 Oxidation state and reversibility**

The results on the oxidative properties of the aged dust are summarised in Figure 10 showing the van Krevelen diagram obtained from the ESI-Orbitrap analysis of a sample collected during experiment $D_2$ (dust control experiments without glyoxal) and a sample collected during experiment $D_{10}$ (dust and glyoxal), both in the dark and with irradiation and 80% RH.

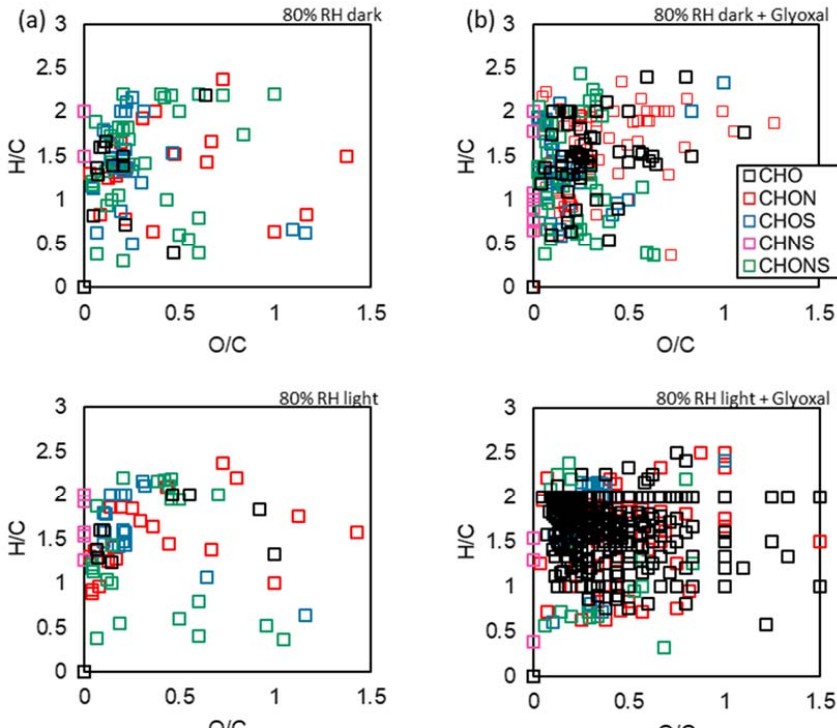

**Figure 10.** Van Krevelen diagrams recorded at 80% RH for: in the top line (a) experiments in the dark for mineral dust only (control experiment $D_2$, left) and one ageing experiment of dust with glyoxal (experiment D10); bottom line (b) same with irradiation.

Under the different conditions presented, the samples exhibited varying levels of particulate organic matter and number of ESI Orbitrap peaks detected.




In the absence of glyoxal, only a few signals are detected. In dark conditions, the
particulate organic matter of filter is 0.9 µg with 102 peaks detected. In irradiated
conditions, the particulate organic matter is 0.8 µg with 86 peaks detected. Peaks are
mostly in the range of O/C < 1 and 0.5 < H/C < 2.5, both in the dark and with irradiation.
The distribution of values of both ratios is rather similar, while the appearance of
molecules for families CHON and CHONS is observed when the lights are on.
The number of detected signals increases significantly in the presence of glyoxal, (right
column of Figure 10), in particular in the presence of light. In dark conditions, the
particulate organic matter on the filter is 1.7 µg, yielding 398 signals detected. In
irradiated conditions, the particulate organic matter is 0.6 µg, resulting in 310 signals
detected. The predominant family in this case is that of CHO molecules. The
appearance of signals with O/C ratio higher than 1 is attributed to photo-oxidation.
For comparison, the O/C ratio of the bulk aerosol measured at 80% RH by the ACSM
and XPS is shown in Figure 11 (results in dry and 30% RH conditions are shown in
Figure S4 in the Supplementary Material).

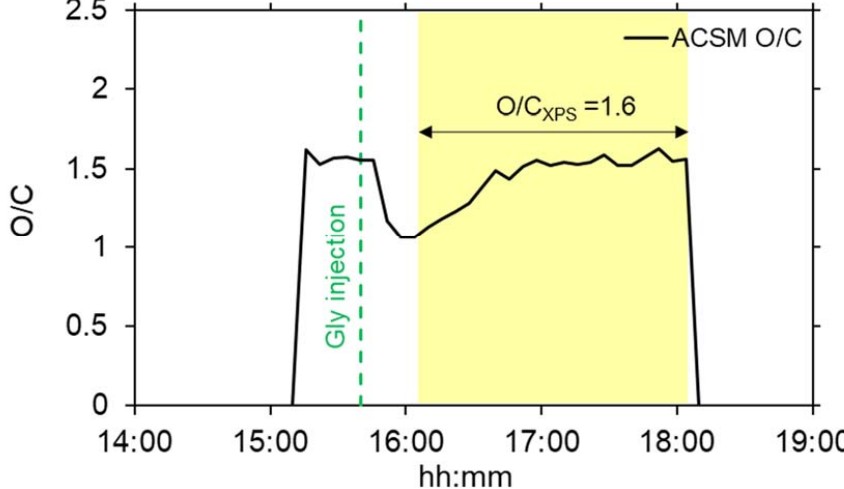


**Figure 11.** Time series of O/C ratio measured with the ACSM during ageing and 80% RH (experiment
$D_{15}$). The black arrows show the duration of filter sampling and the corresponding O/C values obtained
by XPS analysis. The yellow-highlighted portion of the graph indicates the interval where irradiation
takes place, while the green vertical dashed lines indicate the moment of glyoxal injection in the
chamber.



Figure 11 shows that the O/C ratio of the organic material in the native dust is around
1.5. During the uptake of glyoxal, the ratio decreases to 1 to finally revert to 1.5 within
approximately one hour. The agreement between the measurements of the ACSM and
the XPS analysis indicates that the ACSM probes the organic matter at the surface of
the particles, which we expect to be involved in the reactivity towards glyoxal. The
comparison of the results in Figures 10 and 11 suggests that while glyoxal and high
volatility oxidation products evaporate, low volatility and heavy compounds such as
oligomers, remains on the dust and modifies in an irreversible way its surface
composition. This is in agreement with previous studies carried out on ammonium
sulphate seeds (Kroll et al., 2005; Galloway et al., 2009; De Haan et al., 2020; Hu et
al., 2022;).

### 903    4 Conclusive remarks

In this paper, we investigated the uptake of glyoxal by realistic submicron mineral dust
aerosol particles from a natural soil (Gobi Desert), airborne in a large simulation
chamber, and aged under variable experimental conditions of relative humidity,
irradiation, and ozone concentrations. Results can be summarised and commented as
follows:
•    The uptake of glyoxal on the dust particles occurs in humid conditions exceeding

30% RH. These observations agree with the results of Liggio et al. (2005a;

2005b) on the uptake of glyoxal on ammonium sulphate aerosols, observing the

formation of organic matter only when RH exceeded 50%. Trainic et al. (2011)

found that the uptake of glyoxal on glycine and ammonium sulphate particles

occurred only when the relative humidity was above 35%. On the contrary, both

Shen et al (2016) and Zogka et al. (2024) demonstrated that the uptake can occur

in dry conditions too, which we did not observe.

•    The uptake of glyoxal on the dust particles starts as soon as the glyoxal is injected

in the chamber. In this study we used a single and instantaneous injection of

glyoxal, and not a constant steady state flux. Furthermore, in humid conditions,

upon injection, glyoxal is partitioned rapidly between the gas phase and the

chamber walls. Both facts are actually an advantage to scale our results to

ambient conditions. Indeed, Volkamer et al. (2005) estimated that the lifetime of

glyoxal in the daytime is of around 1.3 hours. Alvarado et al. (2020) showed that



the long-range transport of glyoxal produced from a point source (Canadian
wildfires) may be possible only by invoking the progressive oxidation of its longer-
lived precursors in the plume. In the scenario where dust aerosols interact with a
glyoxal plume from a point source, one can expect an interaction time of a few
minutes, compatible with that of this study.
• At 80% RH, the measured uptake coefficient of glyoxal on mineral dust is $\gamma$ = (9
± 5) × $10^{-3}$. Because the uptake follows a first order kinetic, the measured uptake
coefficient is independent on the glyoxal concentration and transferable to
atmospheric conditions. The uptake coefficient on dust is nearly two orders of
magnitude higher than for ammonium sulphate ($\gamma_{gas-AS}$ = 1.1 (± 0.2) × $10^{-4}$) at the
same relative humidity (our study as well as Curry et al., 2018; De Haan et al.,
2020; Galloway et al., 2009; Liggio et al., 2005b, a; Trainic et al., 2011) but lower
than $\gamma_{gas-AS}$ = 2.9 × $10^{-3}$ at a lower relative humidity (Liggio et al., 2005).
• The difference could be due to the higher hygroscopicity of ammonium sulphate,
enhances water's competition with glyoxal for adsorption sites at 80% RH, when
indeed ammonium sulphate is deliquescent. This suggest nonetheless that dust
aerosols could play a very substantial role in the formation of organic aerosols at
high relative humidity compared to ammonium sulphate, which is often used as
an aerosol proxy. Additionally, we found that the uptake coefficient measured by
the loss of gas-phase glyoxal molecules agrees very well with the rate of
formation of secondary organic mass on the particles. This suggests that the
totality of the mass of reacting glyoxal is transformed in organic matter on the
surface of the dust particles.
• The uptake of glyoxal and the formation of organic matter last approximately 20
minutes, after which evaporation occurs. However, we demonstrate the uptake
of glyoxal modifies irreversibly both the composition and the physical properties
of mineral dust. Oligomers and organic acids are detected on the dust even after
the uptake has finished. Our findings support several key insights into the
irreversible uptake of glyoxal on mineral particles, as discussed by Shen et al.
(2016), that found that glyoxal oligomers exhibited a higher degree of
oligomerization (≥ 4) than previously reported in the aqueous phase and on acidic
seed particles (≤ 3; Liggio et al., 2005a; Nozière et al., 2009) and that adsorbed
water on particles favoured the formation of oligomers. This is also in agreement



with the field observations by consistent with the observations conducted by
Wang et al. (2015).
• The presence of organic acids, such as glycolic and glyoxylic acids has
implications for aerosol pH, as they can influence the acidity of aerosols and their
ability to dissolve metals, potentially impacting atmospheric chemistry and the
reactivity of aerosol particles (Giorio et al., 2022). Changes in aerosol pH can, in
turn, affect the hygroscopic properties of aerosols, influencing their ability to
adsorb water and grow in size. Indeed, we observe that the volume of the dust
particle increases during the uptake and that this growth is persistent, henceforth
becoming more efficient in interacting with visible light and form cloud droplets.
The newly formed organic matter from glyoxal on dust particles could also alter
the aerosol's optical properties, affecting its ability to absorb solar radiation, as
recently observed in aqueous solution (De Haan et al., 2023) or on ammonium
sulphate aerosols (De Haan et al. 2020; Trainic et al. 2011). For example, the
presence of hydrated glyoxal oligomeric structures has already been observed to
have UV radiation absorption properties (Kalberer et al., 2004; Shapiro et al.,
2009).

In conclusion, our study reveals a significant quantitative transfer of gas-phase glyoxal
molecules to mineral dust aerosol surfaces, occurring within a timescale of a few
minutes, underscoring the important role of dust-glyoxal interactions in the
atmosphere. Neglecting the uptake pathway on dust could result in an underestimation
of glyoxal removal from the atmosphere, potentially leading to disparities between
model predictions and observed gaseous concentrations of glyoxal (Kluge et al., 2023;
Ling et al., 2020; Volkamer et al., 2007; Washenfelder et al., 2011). The results of this
study could have important implications for the aerosol direct and indirect radiative
effect and aerosol pH. Further studies should investigate dust from different sources
and mineralogy, poly-disperse size distribution including the coarse mode and lower
glyoxal concentration.

***Data availability***. The simulation chamber experiments that support the findings of this study are
available through the Database of Atmospheric Simulation Chamber Studies (DASCS) of the
EUROCHAMP Data Centre (https:// data.eurochamp.org/ data- access/ chamber- experiments/).
**Code availability**. The routine used for fitting the size distribution is available at
https://doi.org/10.5281/zenodo.8135133 (Baldo and Lu, 2023). Note that in this study we only



used the size distribution measured by the OPC instrument, which was fitted with a lognormal
function.
**Author contributions**. PF, JFD and FB conceptualized the study. PF and FB led the paper
writing, with contributions from all the authors. JFD provided with expertise on multi-phase
chemistry. CB analysed the aerosol size distribution data. VM supervised the analysis of PTR-
MS data. FB and CG performed the ESI-Orbitrap analysis of filter samples. FB and JFB
performed the analysis of ACSM observations. FB, TB and AG performed the SFE/CG-MS
analysis of filter samples. FB, GN and SC performed the thermo-optical analysis of filter
samples. FB, MC, AB, EP, VM, BPV and PF conducted the chamber experiments. MR
provided with the soil sample and expertise on heterogeneous chemistry. PF provided with
funding.
**Competing interests**. The authors declare no competing interests.
**Special issue statement**. This article is not part of a special issue. It is not associated with a
conference.
**Acknowledgements**. The AERIS data center (www.aeris–data.fr) is acknowledged for
distributing and curing the data produced by the CESAM chamber through the hosting of the
EUROCHAMP data center (https://data.eurochamp.org). The initial contribution of M. Giordano
(Afri-SET) to the conceptualisation of the project is gratefully acknowledged.
**Financial support**. This work has received funding from the French National Research
Agency (ANR) though the research project CLIMDO under the grant number ANR-19-CE01-
0008-02. It has received support from the European Union's Horizon 2020 research and
innovation program through the EUROCHAMP-2020 Infrastructure Activity under grant
agreement no. 730997. CNRS-INSU is gratefully acknowledged for supporting the CESAM
chamber as a national facility as part of the French ACTRIS Research Infrastructure.



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
