# Peer review of "Formation and composition of organic aerosols from the uptake of glyoxal on natural mineral dust aerosols: a laboratory study"

_EGUsphere, 2024_

## Referee Comment (RC1)

**Review of "Formation and composition of organic aerosols from the uptake of glyoxal on natural mineral dust aerosols: a laboratory study"**

**General comments**

In this study, the authors investigated the heterogeneous reactions of glyoxal on real mineral dust in the CESAM simulation chamber, which is an upgraded vision of the previous studies. Multiple instruments were employed to perform complementary measurements of both gas and particle phases. Key influencing factors were carefully considered, including the presence or absence of irradiation and oxidant ozone, as well as varying humidity conditions ranging from <5% to 80% RH. The chemical composition of reaction products was characterized both online and offline, and uptake coefficients were calculated. This study highlights the substantial uptake of glyoxal by mineral dust and the resulting modifications in particle properties, including hygroscopicity, acidity, optical characteristics, and reactivity. The manuscript is well written. The contribution of this study is within the scope of ACP. There are only a few aspects that require further discussion, and the manuscript would benefit from reorganization, particularly in the results section, to enhance clarity and conciseness. Therefore, I recommend a minor revision before publication.

**Major comments:**

1.  This study is impressive, with comprehensive techniques and results. However, the authors should refine the manuscript to ensure that key points are not overshadowed by excessive detail.

    For instance, while the methodology section contains valuable information, this is not a technical paper. Consider shortening this section and moving some details to Supporting Information (SI) if necessary. Additionally, a table summarizing the complementary functions of the instruments used would greatly enhance clarity.

    In the results section (3.3 Organic Composition), the authors provide a detailed discussion of organic fragments and molecules in the particle phase but lack clear linkages between them. Given the expected diversity of organic products, it is important to highlight which ones are most significant and why. For example, the role of oligomers and organic acids deserves emphasis. Reorganizing this section to focus on key products and scientific questions, rather than structuring it around the instruments used, would improve readability and impact.

2.  Glyoxal concentrations play a potentially import role in the reactions. Why did the authors choose much higher concentration rather than atmospherically relevant values. i.e., 10-100 pptv? Was it due to technical limitations? Please provide an explanation.

3.  Wall loss is a critical factor in accurately calculating the uptake coefficient and warrants further discussion. While its size dependence is addressed in the SI, it is important to clarify whether wall loss also varies with different RH conditions. Additionally, the authors state that the uptake coefficient in humid air is independent of concentration. To support this claim, please provide evidence or relevant references.

4.  The authors did not observe uptake under dry condition. However, given that glyoxal is highly water-soluble—more soluble than methylglyoxal, which can undergo hydration in the gas phase even at <5% RH (Axson et al., 2010). One might expect some uptake to still occur in relatively dry air (though not necessarily at 0% RH) ), but it did not. Can the authors provide potential explanations for this observation? Additionally, a closer look at the volume concentration curve,it shows a slight enhancement after glyoxal injection, though less pronounced than under humid conditions. I recommend exploring alternative methods to better illustrate and analyse the results in dry air.

**Minor comments:**

Line 116: Give full name of OA.

Line 127: Give full name of CESAM or add some explanation, e.g, the abbreviation in French for Experimental Multiphasic Atmospheric Simulation Chamber.

Line 181: Please show the FTIR spectra, e.g., in SI.

Line 457: Fig 1 a) RH=0%, is it real 0%? Or some value < 5% which is the lowest RH listed in table1. Please revise. Same comment for Fig. S4.1.

Line 485-486: Consider putting the result in presence of ozone in the SI.

Line 502: It is not clear why Fig 2 can confirm the statements above. Please reshape.

Line 573: Fig 4, please add the legend for the red and blue curves.

Line 599: The update and formation of OA process is reversible.

Line 610: Table 2: Do not use the symbol γ for OA formation. Consider using e.g., Y (yield) instead.

Line 625: Shen et al. (2016) also used $\alpha$-$Al_2O_3$. Please add it.

Line 627: How long was the exposure time?

Line 631: Change "$AlO_3$" to "$\alpha$-$Al_2O_3$".

Line 635: What are the values of update coefficients with BET surface area? Please calculate and compare them with the corresponding literature.

Line 685: Change 28 m/z to m/z 28. Same for 44 m/z and the other fragments. Please go through the whole manuscript and revise accordingly.

Line 688-690: Need more explanation for this observation.

Line 783: Do not use "seem to". Please change to another more appropriate word.

Line 929-936: Please add the comparison of uptake coefficients with Shen et al. (2016).

References:

Axson, J. L., Takahashi, K., De Haan, D. O., and Vaida, V.: Gas-phase water-mediated equilibrium between methylglyoxal and its geminal diol, Proc. Natl. Acad. Sci. U.S.A., 107, 6687–6692, https://doi.org/10.1073/pnas.0912121107, 2010.

---

## Author Comment (AC1)

**Reply to referee comments (RC) to manuscript egusphere- 2024-4073**

The authors are grateful to anonymous reviewer 1 and 2 for providing very valuable feedbacks to the manuscript. Our detailed point-by-point response to their comments are reported below.

Both referees, implicitly or explicitly, indicated that the manuscript was presenting too many details, which were hampering the understanding the main conclusions and novelties of the research. To overcome this issue, we have reorganized the manuscript as follows

- A significant part of the experimental section is now reported in the supplementary material, only the major details remain in the main text
- The "results and discussion" section has been separated to highlight the implications of the results
- The detailed discussion of the mass spectra in the composition section has been moved in the supplementary material, to highlight the discussion on the formation of acids and oligomers

We believe that these significant modifications improved the reading and the understanding of the manuscript, fostering its significance.

**Reply to Referee # 1 comments (RC) to manuscript egusphere- 2024-4073**

We thank Referee #1 for the complete and thoughtful revision of our manuscript.

In the following the Referees' remarks and questions are in black and our replies are in light blue

**Referee Comments #1**

**General comments**

In this study, the authors investigated the heterogeneous reactions of glyoxal on real mineral dust in the CESAM simulation chamber, which is an upgraded vision of the previous studies. Multiple instruments were employed to perform complementary measurements of both gas and particle phases. Key influencing factors were carefully considered, including the presence or absence of irradiation and oxidant ozone, as well as varying humidity conditions ranging from <5% to 80% RH. The chemical composition of reaction products was characterized both online and offline, and uptake coefficients were calculated. This study highlights the substantial uptake of glyoxal by mineral dust and the resulting modifications in particle properties, including hygroscopicity, acidity, optical characteristics, and reactivity. The manuscript is well written. The contribution of this study is within the scope of ACP. There are only a few aspects that require further discussion, and the manuscript would benefit from reorganization, particularly in the results section, to enhance clarity and conciseness. Therefore, I recommend a minor revision before publication.

**Major comments:**

1. This study is impressive, with comprehensive techniques and results. However, the authors should refine the manuscript to ensure that key points are not overshadowed by excessive detail. For instance, while the methodology section contains valuable information, this is not a technical paper. Consider shortening this section and moving some details to Supporting Information (SI) if necessary. Additionally, a table summarizing the complementary functions of the instruments used would greatly enhance clarity.

Following your suggestion, we revised and shortened the methodology section in the main text (Section 2) and reported details in the supplementary material (Text S1 and S2). Additionally, we have included Table S1 in the Supplementary Information, which summarizes the complementary functions of the instruments used, to improve clarity and accessibility of the experimental setup.

In the results section (3.3 Organic Composition), the authors provide a detailed discussion of organic fragments and molecules in the particle phase but lack clear linkages between them. Given the expected diversity of organic products, it is important to highlight which ones are most significant and why. For example, the role of oligomers and organic acids deserves emphasis. Reorganizing this section to focus on key products and scientific questions, rather than structuring it around the instruments used, would improve readability and impact.

We have followed your recommendation and revised the text accordingly. We have reorganized the discussion in Section 3.4, placing greater emphasis on the most significant organic products—particularly oligomers and organic acids—and their roles in the particle phase. The section has been reorganized around key findings and scientific questions, rather than by instrumentation, to enhance readability and highlight the main outcomes of the study. Additional material, including the presentation of the mass spectra is now reported in Text S7 in the supplementary material.

2. Glyoxal concentrations play a potentially import role in the reactions. Why did the authors choose much higher concentration rather than atmospherically relevant values. i.e., 10-100 pptv? Was it due to technical limitations? Please provide an explanation.

The decision to use higher glyoxal concentrations, rather than atmospherically relevant values of 10-100 pptv, was motivated by the need to overcome the rapid wall loss of glyoxal on the chamber walls in humid conditions, when glyoxal is more likely to be adsorbed onto the chamber walls, leading to a reduction in the concentrations available for interaction with the aerosol particles. The rates of the loss of glyoxal on the chamber walls were quantified by dedicated experiments measuring the wall loss rate constant, which we now report in Text S3 of the Supplementary Material. These consisted in injecting 1 ppm of glyoxal into the chamber and then introducing water vapor to test different relative humidity (RH) conditions. The wall loss constant in dry conditions ($k_{WL,Dry}$) was found to be 4 ($\pm$ 2)$\cdot 10^{-5}$ $s^{-1}$, and roughly two orders of magnitude in humid conditions ($k_{WL,Humid}$; 1.8 ($\pm$ 0.9)$\cdot 10^{-3}$ $s^{-1}$. Such increase in concentration was aimedto ensure the measurements of the organic aerosol composition, with both online and offline techniques.

3. Wall loss is a critical factor in accurately calculating the uptake coefficient and warrants further discussion. While its size dependence is addressed in the SI, it is important to clarify whether wall loss also varies with different RH conditions. Additionally, the authors

state that the uptake coefficient in humid air is independent of concentration. To support this claim, please provide evidence or relevant references.

We thank Reviewer #1 for those comments, and, as also discussed in the previous answer, we agree with the statements indicating the importance of the mechanisms of wall loss, both for dust particles and the gas-phase. For sake of clarity, we address those separately.

- The wall losses for dust particles are corrected as presented in the Text S4 in the supplementary material. These are based on the deposition model by Lai and Nazaroff (2000), which provides a theoretical framework for estimating particle lifetime inside a reactor. The model assumes a one-dimensional vertical flux and neglects particle rebounds from the walls, which act as a perfect sink removing particles upon impaction. These conditions are not completely met in the CESAM chamber as a consequence of two phenomena: the presence of an internal fan operated to enhance mixing, inducing an additional horizontal movement to the particles, and physical properties of dust particles, which are electrostatically active and can rebound from the walls. As discussed in Battaglia et al. (2025), in dry conditions, the size-average rate of deposition loss rate coefficient rate is 2.5 (±0.8) $10^{-5}$ $s^{-1}$, resulting in calculated particle lifetime between 9 and 11 h, significantly lower than the expected from the theoretical calculations, indicating a particle lifetime of the order of 24h (see Alfarra et al. 2023a). In the presence of water vapor, the average lifetime reduced to 2-3 hours (not shown in Battaglia et al., 2025).

- The wall loss of the gas phase is certainly depending on RH. This dependence was considered by developing a specific kinetic model, which is now described in Text S3 in the supplementary material. Regarding the losses (uptake) of glyoxal on dust particles, Reviewer #1 notes our sentence saying "*results indicate that natural Gobi dust is an effective sink of glyoxal, with initial uptake coefficient independent of glyoxal concentration, pointing to a first order removal process...*", Indeed, our experimental results of glyoxal uptake on mineral dust could be described with a first-order kinetic model, which assumes that the uptake rate is independent of the initial concentration of the gas-phase reagent. Similar first-order uptake behavior has been reported in previous studies across a range of particle substrates. Shen et al. (2016) studied glyoxal uptake on clean and $SO_2$-aged mineral particles; Zogka et al. (2022) investigated uptake on a wide variety of natural dusts and mineral proxies; Go et al. (2018) examined reactions on methylammonium-containing inorganic salts; and Liggio et al. (2005) focused on glyoxal uptake onto ammonium sulfate particles. Despite the diversity of substrates, all these studies support a first-order uptake process, reinforcing our interpretation of the glyoxal uptake mechanism under the experimental conditions presented in this work.

4. The authors did not observe uptake under dry condition. However, given that glyoxal is highly water-soluble—more soluble than methylglyoxal, which can undergo hydration in the gas phase even at <5% RH (Axson et al., 2010). One might expect some uptake to still occur in relatively dry air (though not necessarily at 0% RH) but it did not. Can the authors provide potential explanations for this observation? Additionally, a closer look at

the volume concentration curve, it shows a slight enhancement after glyoxal injection, though less pronounced than under humid conditions. I recommend exploring alternative methods to better illustrate and analyze the results in dry air.

We thank Referee #1 for verifying the observation in such a detail. Indeed, based on the results of Shen et al. (2016) and Zogka et al. (2024) we expected some uptake in dry conditions too. While it is possible that the light enhancement in the volume concentration curve might indicate that some uptake is occurring, this and the concentration of the organic aerosol formed are below the detection limit of our experimental techniques.

The absence of glyoxal uptake under dry conditions in our experiments could be due to the limited reactive surface area available for heterogeneous interactions. In contrast to our study, Shen et al. (2016) and Zogka et al. (2024) examined glyoxal uptake on deposited soils, with a significantly higher availability of reactive sites. Shen et al. (2016) used model powders with BET (Brunauer-Emmett-Teller) surface areas ranging between 1.4 and 440 $m^2$ $g^{-1}$, respectively. In the conservative assumption that only 5 mg of the model powder was used, the total reactive surface area was up to 2.2 $m^2$. Zogka et al. (2024) reported that the BET surface areas of the Gobi soil used in their experiments was 10.5 ± 1.0 $m^2$ $g^{-1}$. Using even just 1 g would yield a surface area of the order of 10.5 $m^2$.

On the other hand, in our study we privileged the realism with respect to the dust conditions in the atmosphere. The generated aerosols were distributed in a single mode centered around 300 nm (mobility diameter). This approach inherently limits the total available surface area. The aerosol surface area calculated from the size distribution measurements across our experiments was in the range 0.35-6.3 × $10^{-3}$ $m^2$ $m^{-3}$. The corresponding total surface area ranged from approximately 0.002 to 0.026 $m^2$ (the CESAM chamber volume is 4.2 $m^3$), several orders of magnitude lower than in the experiments by Shen et al. (2016) and Zogka et al. (2024). This discussion is now added in the main text of the revised manuscript (section 3.2).

**Minor comments:**

Line 116: Give full name of OA.

We have corrected the text accordingly by providing the full name of OA (organic aerosol) at line 116.

Line 127: Give full name of CESAM or add some explanation, e.g, the abbreviation in French for Experimental Multiphase Atmospheric Simulation Chamber.

We have revised the text at line 127 to include the full name of CESAM, specifying that it stands for "Chambre Expérimentale de Simulation Atmosphérique Multiphasique", the French abbreviation for Experimental Multiphasic Atmospheric Simulation Chamber.

Line 181: Please show the FTIR spectra, e.g., in SI.

We have now included the FTIR spectra in Figure S1 of the Supplementary Material.

Line 457: Fig 1 a) RH=0%, is it real 0%? Or some value < 5% which is the lowest RH listed in table1. Please revise. Same comment for Fig. S4.1.

The RH measurements in our study were conducted using the Vaisala® HMP234 transmitter, which is designed for accurate humidity measurements across a full range of

0–100% RH. According to the manufacturer's specifications, the HMP234 is capable of measuring RH values down to 0%.

In the context of our experiments, the "RH = 0%" condition refers to measurements taken in a dry environment where the RH was effectively at or near zero. Given the capabilities of the HMP234, we are confident in the accuracy of these low RH measurements.

Line 485-486: Consider putting the result in presence of ozone in the SI.

In response, we have moved the results in the presence of ozone to the Supplementary Information and integrated them intoFigure S2. This figure presents the timeline of ageing experiment $D_{14}$, where submicron dust was exposed to gas-phase glyoxal under humid conditions (78% RH) at 293 K and 1450 ppbv of ozone.

Line 502: It is not clear why Fig 2 can confirm the statements above. Please reshape.

This is now discussed in Section 3.4

Line 573: Fig 4, please add the legend for the red and blue curves.

Done

Line 610: Table 2: Do not use the symbol γ for OA formation. Consider using e.g., Y (yield) instead.

As a matter of fact, we do not report a yield of OA formation but rather a rate. Therefore, for consistency with the gas phase, we prefer to keep using the symbol γ. We have revised the text to clarify.

Line 625: Shen et al. (2016) also used α-Al2O3. Please add it.

Done

Line 627: How long was the exposure time?

The exposure time of glyoxal to mineral dust, corresponding to the period between the glyoxal injection and the stabilization of aerosol mass concentration and chemical composition, was of the order 5 to 10 minutes. As also detailed in response to one of the minor comments of Reviewer #2, this short interaction time is since in our experiments we performed a single injection of glyoxal, which, due to its high solubility and reactivity, rapidly partitioned to both the aerosol surface and the chamber walls.

Our exposure time is realistic of atmospheric conditions, and shorter than previously reported in the literature. Shen et al. (2016) exposed mineral dust to glyoxal for approximately 30 minutes (see Section 2.3.2 of the revised manuscript). The experiments reported by Zogka et al. (2024) used an exposure time between 40 to 60 minutes, and reached a quasi-steady-state uptake (see Sections 2.4.3 and 2.4.4).

Line 631: Change "AlO3" to "α-Al2O3".

Done

Line 685: Change 28 m/z to m/z 28. Same for 44 m/z and the other fragments. Please go through the whole manuscript and revise accordingly.

This is now done

Line 783: Do not use "seem to". Please change to another more appropriate word.

We have revised the sentence, accordingly changing it from "Light and the presence of ozone seem to favor its formation" to "Light and the presence of ozone favor its formation" at line 830 in the revised manuscript.

Line 929-936: Please add the comparison of uptake coefficients with Shen et al. (2016).

This is now done

**References**

Alfarra, R., U. Baltensperger, D. M. Bell, S. G. Danelli, C. Di Biagio, J.-F. Doussin, P. Formenti, M. Gysel-Beer, D. Massabo, G. McFiggans, et al. 2023a. Preparation of the experiment: Addition of particles. In A practical guide to atmos. simulation chambers, ed. J.-F. Doussin, H. Fuchs, A. Kiendler-Scharr, P. Seakins, and J. Wenger, 163–206. Cham: Springer International Publishing.

Battaglia, F., Baldo, C., Cazaunau, M., Bergé, A., Pangui, E., Picquet-Varrault, B., Doussin, J.-F., & Formenti, P. (2025). Protocol for generating realistic submicron mono-dispersed mineral dust particles in simulation chambers and laboratory experiments. Aerosol Science and Technology, 59(3), 357–369. https://doi.org/10.1080/02786826.2024.2442518

Go, B. R., Gen, M., Chu, Y., & Chan, C. K. (2019). Reactive Uptake of Glyoxal by Methylaminium-Containing Salts as a Function of Relative Humidity. ACS Earth and Space Chemistry, 3(2), 150–157. https://doi.org/10.1021/acsearthspacechem.8b00154

Lai, A. C., & Nazaroff, W. W. (2000). Modeling indoor particle deposition from turbulent flow onto smooth surfaces. Journal of aerosol science, 31(4), 463-476.

Liggio, J., Li, S., & McLaren, R. (2005). Reactive uptake of glyoxal by particulate matter. Journal of Geophysical Research: Atmospheres, 110(D10), 2004JD005113. https://doi.org/10.1029/2004JD005113

Loza, C. L., Chan, A. W., Galloway, M. M., Keutsch, F. N., Flagan, R. C., & Seinfeld, J. H. (2010). Characterization of vapor wall loss in laboratory chambers. Environmental science & technology, 44(13), 5074-5078.

Shen, X., Wu, H., Zhao, Y., Huang, D., Huang, L., & Chen, Z. (2016). Heterogeneous reactions of glyoxal on mineral particles: A new avenue for oligomers and organosulfate formation. Atmospheric Environment, 131, 133–140. https://doi.org/10.1016/j.atmosenv.2016.01.048

Zogka, A. G., Lostier, A., Papadimitriou, V. C., Thevenet, F., Formenti, P., Rossi, M. J., Chen, H., & Romanias, M. N. (2024). Unraveling the Uptake of Glyoxal on a Diversity of Natural Dusts and Surrogates: Linking Dust Composition to Glyoxal Uptake and Estimation of Atmospheric Lifetimes.

---

## Author Comment (AC2)

**Reply to referee comments (RC) to manuscript egusphere- 2024-4073**

The authors are grateful to anonymous reviewer 1 and 2 for providing very valuable feedbacks to the manuscript. Our detailed point-by-point response to their comments are reported below.

Both referees, implicitly or explicitly, indicated that the manuscript was presenting too many details, which were hampering the understanding the main conclusions and novelties of the research. To overcome this issue, we have reorganized the manuscript as follows

- A significant part of the experimental section is now reported in the supplementary material, only the major details remain in the main text
- The "results and discussion" section has been separated to highlight the implications of the results
- The detailed discussion of the mass spectra in the composition section has been moved in the supplementary material, to highlight the discussion on the formation of acids and oligomers

We believe that these significant modifications improved the reading and the understanding of the manuscript, fostering its significance.

**Reply to referee comments (RC) to manuscript egusphere- 2024-4073**

We thank Referee #2 for the complete and thoughtful revision of our manuscript.

In the following the Referees's remarks and questions are in black and our replies are in light blue

**Referee Comments #2**

**General comments:**

This manuscript presents a well-designed laboratory study investigating the uptake of glyoxal on natural mineral dust aerosols from the Gobi Desert. The authors utilize a large simulation chamber to explore the formation of organic aerosols (OA) under varying atmospheric conditions, including different relative humidity (RH) levels and the presence of ozone. The study provides important experimental constraints on glyoxal uptake coefficients and highlights the role of mineral dust in heterogeneous atmospheric chemistry.

Major Comments:

**1. Scientific Significance and Novelty**

The study provides valuable insights into the uptake of glyoxal on mineral dust, yet the novelty should be more explicitly stated. How does this work go beyond previous studies such as Shen et al. (2016) or Zogka et al. (2024)? A clearer articulation of the key new findings would strengthen the introduction.

To the best of our knowledge, our paper reports on the first ever study of the interactions of natural mineral dust aerosols with volatile organic compounds, and glyoxal in particular, following the recommendation of the review paper by Tang et al. (2017). All previous studies in the literature investigated soils or soil surrogates like synthetic minerals, while our study investigate realistic airborne aerosol particles generated from a natural native soil. Our experiments considered submicron mineral dust aerosol particles generated from a natural soil (Gobi Desert) in realistic concentrations, composition and size distribution. While airborne, the dust aerosol was aged under variable conditions of relative humidity, irradiation, and ozone concentrations. Beside the realism of the tools and the experimental conditions, our study investigates, for the first time, both the rate of uptake of glyoxal and the rate of formation of the organic aerosol from the gas-phase uptake, the chemical composition of the particle formed on the dust particles, and finally the implications on the particle microphysics.

There are also several novelties with respect to previous studies on glyoxal by Shen et al. (2016) and Zogka et al. (2024). With respect to Shen et al. (2016), we extend the results to the effects on particle size and provide with the evaluation of both the coefficients of gaseous uptake and formation of the organic matter. With respect to Zogka et al. (2024), who only investigate the uptake, we provide with the investigation of the rate of particle formation, its mechanism, and the complete characterization of the organic composition formed in the particle phase, including the molecular scale. To explicit these points, we have added the reference to Zogka et al (2024) in the introduction and added some lines of text to highlight atmospheric relevance (lines 114-122).

Additionally, we modified the conclusions with the following sentences (lines 763-776) "*In this paper, we investigated the interaction between gas-phase glyoxal and mineral dust. By taking advantage of the capabilities of the CESAM atmospheric simulation chamber to perform multiphase experiments on time scales relevant to atmospheric processes and dispersion, including aerosols, our study extends the previous findings of Shen et al. (2016) and Zogka et al. (2024), who explored the uptake of glyoxal onto soils or soil surrogates like synthetic minerals. Our experiments considered submicron mineral dust aerosol particles generated from a natural soil (Gobi Desert) in realistic concentrations, composition and size distribution. While airborne, the dust aerosol was aged under variable conditions of RH, irradiation, and ozone concentrations. Beside the realism of the tools and the experimental conditions, our study investigates, for the first time, both the rate of uptake of glyoxal and the rate of formation of the organic aerosol from the gas-phase uptake, the chemical composition of the particle formed on the dust particles, and finally the consequences on the particle microphysics*".

**2. Mechanistic Understanding of Organic Aerosol Formation**

The paper suggests that organic aerosol (OA) formation from glyoxal uptake on dust is significant, yet the reversibility of the process raises questions about its atmospheric relevance. Can the authors discuss potential mechanisms for OA stabilization under ambient conditions (e.g., secondary reactions, coagulation)?

Indeed our observations suggest that the uptake of glyoxal on dust exhibits a certain degree of reversibility, as the more volatile compounds, forming the majority of the organic mass concentration, revert to the gas-phase during the reaction, likely by evaporation. However, our findings also clearly indicate that the process modify in an irreversible way

the composition and the size distribution of the mineral dust particles. Molecular analysis by ESI-Orbitrap and SFE/GC-MS revealed a variety of glyoxal-derived compounds, such as glycolic acid, glyoxylic acid, oxalic acid, and several oligomers (e.g., $C_6H_6O_6$, $C_8H_{16}O_{12}$, $C_{10}H_{12}O_{11}$), on the mineral dust particles, particularly at irradiated conditions. After the reaction, the dust size distribution grows permanently to larger sizes.

These observations support the idea that although some glyoxal and its characteristic fragments may volatilize or degrade over time, the formation of lower-volatility, higher molecular weight compounds through secondary reactions—potentially including photochemistry, radical-driven oxidation, and condensation—leads to partial but significant stabilization of the organic aerosol (OA). The influence of ambient factors such as light and ozone further suggests that multiple atmospheric pathways may contribute to this process.

The stabilization of OA products depends in large part on their volatility. The formation of lower-volatility compounds through secondary processes—such as oxidation, photochemical aging, and oligomerization—contributes to their atmospheric persistence. In particular, polar oxygenated products formed from glyoxal are likely to exhibit reduced volatility and can further interact with the dust surface via hydrogen bonding, van der Waals forces, or surface complexation mechanisms.

The nature of the mineral surface and the presence of adsorbed water are critical in facilitating these interactions. Mineral surfaces often provide reactive sites (e.g., hydroxyl groups, metal cations) that can stabilize polar organics through specific interactions. Under humid conditions, a thin layer of adsorbed water may enhance glyoxal uptake and retention by promoting aqueous-phase reactions and acting as a medium for molecular diffusion and stabilization.

This water layer can also help solubilize polar products and mediate their association with surface sites, effectively reducing their volatility and enhancing their persistence. Indeed, relative humidity plays a key role in the effectiveness of the interaction. Under humid conditions, glyoxal exists predominantly in hydrated forms, which are both less volatile and more water-soluble. Using the Myrdal and Yalkowsky (1997) method to estimate saturation vapor pressures at 298 K, each hydration step of glyoxal reduces its vapor pressure dramatically: from $10^{-0.7}$ atm for glyoxal, to $10^{-4.2}$ atm for the mono-hydrate, and $10^{-8.1}$ atm for the dihydrate. These decreases in volatility strongly favor SOA formation. In parallel, the aqueous solubility of hydrated glyoxal forms is greatly enhanced, with effective Henry's law constants of $10^{8.81}$ and $10^{14.09}$ M atm$^{-1}$ for the mono- and dihydrated forms, respectively (Raventos-Duran et al., 2010), compared to $10^{5.56}$ M atm$^{-1}$ for the non-hydrated form (Zhou and Mopper, 1990). These properties promote partitioning into the condensed phase, further supporting the formation of persistent organic material.

**3. Uncertainty in Uptake Coefficients**

The reported uptake coefficient ($\gamma = 9 \times 10\text{-}3 \pm 5$) shows large variability. Given the importance of $\gamma$ in atmospheric modeling, can the authors provide a sensitivity analysis on key parameters (e.g., dust surface area, chamber conditions)? How do wall losses impact the measured $\gamma$ values?

We appreciate the reviewer's insightful comment regarding the variability of the reported uptake coefficient (γ) and the potential implications for atmospheric modeling. In response, we provide below a structured discussion addressing the main factors contributing to this variability, namely: (1) differences in aerosol surface area across experiments, (2) challenges in achieving reproducible glyoxal injections due to significant wall losses, and (3) the influence of chamber wall conditions on glyoxal availability and wall loss kinetics.

1) Variability in aerosol surface area

We acknowledge that the uptake coefficient γ displays some variability across our experiments. A significant factor contributing to this is the variability in the injected aerosol surface area (AS), discussed in Battaglia et al. (2025). In that study, we describe the challenges associated with generating a monodisperse mineral dust aerosol with a mobility diameter centered at approximately 300 nm. This specific size range was chosen to ensure a sufficiently long particle lifetime in the CESAM chamber, which is essential for conducting heterogeneous uptake experiments with glyoxal. However, achieving this size distribution in a consistent and reproducible manner is technically demanding.

As a result, the total injected surface area varied between experiments from $3.5 \times 10^{-4}$ to $6.3 \times 10^{-3}$ m² m⁻³. This variability directly affects the value of γ, as the uptake coefficient is normalized by the available particle surface area.

2) Variability in glyoxal injection due to wall losses

Another important source of variability comes from the initial concentration of gas-phase glyoxal, which in our experiments varied between 370 and 800 ppbv. Despite efforts to standardize the injection protocol, such variability is unavoidable. Injecting glyoxal in a reproducible manner is also challenging due to its strong tendency to undergo wall loss, especially under humid conditions. These losses occur rapidly and are influenced by environmental factors such as temperature and RH, as well as by the physicochemical interactions between glyoxal and chamber walls. As consequence, the amount of glyoxal available for uptake at the particle surface can differ from one experiment to another. The kinetics of these processes, and our strategy to quantify them, are now discussed in detail in Text S2 of the Supplementary Material.

3) Influence of chamber wall condition on wall loss kinetics and glyoxal availability

Finally, an additional source of variability arises from the condition of the CESAM chamber walls themselves. The kinetics of glyoxal wall loss are not only strongly dependent on RH, but also on the surface state of the chamber—e.g., the degree of previous exposure to glyoxal, dust, ozone, or other reactive gases. The surface "history" of the chamber, including the point in time during an experimental campaign when the uptake experiment is performed, can significantly affect wall reactivity and thus the gas-phase concentration of glyoxal available to react with aerosols.

In conclusion, we recognize the importance of γ for atmospheric modeling and agree that its interpretation must account for experimental uncertainties. Together, the factors discussed explain the observed range of uptake coefficients and highlight the complexity of simulating real-world heterogeneous processes under controlled laboratory conditions.

**4. Role of Relative Humidity and Surface Chemistry**

The data indicate a strong dependence of glyoxal uptake on relative humidity (RH), but the underlying reasons are not fully explored. Is this due to enhanced solubility, surface adsorption, or heterogeneous reactions? Additional discussion on the molecular-level processes involved would improve clarity.

The enhanced uptake at higher RH can be attributed to several interrelated factors. First, increased RH facilitates the formation of thin water films or hydrated layers on the dust surface, which enhances glyoxal's solubility and promotes its partitioning into the aqueous phase. This aqueous environment not only allows for more efficient uptake but also creates a medium for subsequent aqueous-phase reactions.

Second, RH likely plays a key role in promoting surface-mediated and heterogeneous chemistry. The water layer acts as a solvent, enabling hydration, oxidation, and oligomerization of glyoxal—processes that are significantly less favorable under dry conditions. This is supported by our molecular analysis, which revealed the presence of a variety of oxidation products and oligomers (e.g., glycolic acid, oxalic acid, and higher-order oligomers like $C_8H_{16}O_{12}$) predominantly at high RH and under irradiation. The diversity and persistence of these products point to active aqueous-phase and surface-bound chemistry facilitated by RH.

Together, these findings suggest that RH enhances glyoxal uptake not only by increasing solubility but also by enabling a suite of heterogeneous and aqueous-phase reactions that stabilize a portion of the organic mass on the dust surface. We have clarified this point further in the revised manuscript (section 4.2).

In addition, the hydrated forms of glyoxal are significantly more polar and thus more likely to adhere to the mineral surface through hydrogen bonding or electrostatic interactions. These interactions contribute to the stabilization of glyoxal and its reaction products on the surface, reducing their volatility and favoring the formation of secondary organic aerosol.

As already discussed in our response to comment 2, hydration of glyoxal dramatically lowers its volatility—by several orders of magnitude—and increases its solubility and polarity, enhancing its ability to interact with polar sites on the mineral dust surface. These molecular characteristics reinforce the idea that RH not only increases glyoxal uptake but also promotes its stabilization on dust through strong surface interactions.

Together, these findings suggest that RH enhances glyoxal uptake not only by increasing solubility but also by enabling a suite of heterogeneous and aqueous-phase reactions and surface interactions that stabilize a portion of the organic mass on the dust surface. We have clarified this point further in the revised manuscript.

**5. Optical and Hygroscopic Property Changes**

The study mentions that glyoxal uptake modifies dust properties, yet quantitative data on optical and hygroscopic changes are limited. Can the authors provide experimental evidence (e.g., changes in scattering, hygroscopic growth factors) to support these claims?

The current manuscript focuses on the formation and the chemical composition of the organic aerosols on the dust particles, and provides insights on the kinetic rates, and presence of low- and high-volatility compounds on the surface of the particle dust, in particular acids and oligomers which could alter the particle's pH. The paper also shows that the formation results in the irreversible growth of the particle size, which we attribute to the enhancement of dust's hygroscopicity, and which should result in the change of the dust efficiency in scattering and absorbing the solar radiation. While the formal analysis of those data is beyond the scope of this paper, we have added a specific Section 4.3 to address the Referee's comments. A manuscript is currently being prepared to present concurrent measurements of hygroscopic growth factors and scattering and absorption efficiencies.

**Minor Comments:**

1. Experimental Controls and Reproducibility

While the authors conducted control experiments with ammonium sulfate, it is unclear whether repeated experiments were performed to assess reproducibility. Were there significant variations across trials, and how were they addressed?

The issue of reproducibility was significantly considered. Table 1 reports several duplicates and triplicates of experiments performed in same nominal conditions, and this is just a selection of the many more trial experiments which we performed in the laboratory to understand the chemical system that we are dealing with, and its variability. As discussed in the answer to comment #3, a large degree of variability is indeed something that we have to live with as a result of the intrinsic experimental challenges provided by both the glyoxal and the mineral dust, which are atmospherically short-lived and reactive compounds. In this respect too, our experiments are extremely realistic of the interaction that could occur in the atmosphere, and its variability due to the environmental conditions that air masses can explore.

2. Data Interpretation in Figure 1 and Figure 2

The time series data suggest transient organic aerosol formation, followed by evaporation. Could this be due to chamber dilution effects rather than desorption? A more detailed discussion would clarify the interpretation.

We thank the reviewer for raising this important point. The transient organic aerosol formation followed by apparent evaporation was carefully analyzed to distinguish real chemical processes from physical effects such as chamber dilution.

To address this, we applied a well-established dilution correction method as detailed in Text S3.1 of the Supplementary Material. This correction accounts for the dilution of gas-phase species and aerosol particles caused by the injection of particle-free nitrogen flow needed to balance the sampling flow and maintain constant chamber pressure.

In addition, size-dependent particle losses due to interactions with chamber walls and sampling lines (wall losses) were quantified and corrected for as described in Text S3.2. By combining these corrections, the reported changes in aerosol concentration and composition accurately reflect chemical uptake and transformation rather than dilution.

Therefore, the observed transient organic aerosol behavior cannot be attributed only to dilution effects but is consistent with partial reversibility of glyoxal uptake and chemical processing on mineral dust. This conclusion is further supported by molecular analyses showing the formation of less volatile glyoxal-derived oligomers and oxidation products.

In summary, dilution and wall losses were fully accounted for, and the temporal trends in Figures 1 and 2 reflect genuine heterogeneous chemistry rather than experimental artifacts.

3. Chemical Characterization of Oligomers

The study identifies C4–C10 oligomers but does not provide detailed structural assignments. Were tandem MS (MS/MS) or other advanced techniques used to confirm molecular identities? If not, adding structural information would strengthen the findings.

We thank the reviewer for this important observation. While tandem MS (MS/MS) experiments were not performed in this study, the assignment of molecular formulae was conducted using high-resolution Orbitrap mass spectrometry (mass resolution = 100,000 at m/z 400) with a stringent peak assignment protocol (Zielinski et al., 2018) allowing for mass errors within ±4 ppm. This high mass accuracy, combined with specific elemental constraints (e.g., H/C, O/C, N/C, S/C ratios, nitrogen rule), ensures a high level of confidence in the formula assignment and supports the robustness of our chemical characterization.

Additionally, we emphasize that molecular formulas attributed to glyoxal chemistry (e.g., hydration, oxidation, and oligomerization products) were observed exclusively in the presence of both mineral dust and glyoxal. To further validate these assignments, we conducted a series of blank experiments with dust aerosol under identical chamber conditions but without glyoxal injection. In those control experiments, no glyoxal-related molecular formulas were detected, strongly supporting the attribution of these compounds to heterogeneous chemistry between glyoxal and the mineral surface.

Thus, although we acknowledge the absence of direct structural elucidation via MS/MS or NMR, the high reliability of the molecular formula assignments—combined with the fact that these species were observed only in the co-presence of glyoxal and dust—provides strong support for the proposed structures. In addition, the assignation is consistent with established glyoxal chemistry, particularly under humid conditions, where hydration promotes lower volatility and enhanced interactions with mineral surfaces. We have added a specific section to discuss the structural assignments and chemical mechanism (Section 4.2)

4. Comparison with Atmospheric Conditions

The chamber experiments are performed at controlled conditions, but how do these findings translate to real atmospheric environments? Discussion on differences in dust properties, glyoxal concentrations, and timescales in the ambient atmosphere would be beneficial.

While our experiments were conducted in a controlled chamber environment, multiple aspects of the study were designed to reflect atmospherically relevant conditions and processes.

First, the uptake of glyoxal on mineral dust was observed to occur immediately after glyoxal injection into the chamber. In this study, we employed a single, instantaneous glyoxal injection rather than a continuous flux. This design choice, combined with the rapid partitioning of glyoxal to both particles and chamber walls under humid conditions, mimics the transient interaction conditions found in the atmosphere.

Indeed, as shown by Volkamer et al. (2005), the typical atmospheric lifetime of glyoxal is on the order of 1.3 h during the day, while Alvarado et al. (2020) demonstrated that long-range transport of glyoxal is unlikely unless secondary production from longer-lived precursors is invoked. In the real world, the interaction of mineral dust with glyoxal laden-air masses from biomass burning or urban pollution, will take place within a few minutes (ref) , comparable to the timescale explored in our chamber experiments.

In addition, although the initial glyoxal mixing ratios used in our study (hundreds of ppbv) are higher than typical background atmospheric levels (generally 10–100 pptv, Fu et al., 2008), this is a common and necessary feature of chamber studies designed to yield quantifiable signals within limited experimental durations. It's important to note that due to rapid wall partitioning under humid conditions, the effective concentration of glyoxal available for heterogeneous uptake is significantly lower than the initial value. This loss suggests that the actual exposure of aerosols to glyoxal in the chamber may not be far from ambient conditions.

Moreover, the mineral dust aerosol used in this study was size-selected around 300 nm in diameter—an atmospheric size mode representative of particles that are efficiently transported over long distances, enhancing the atmospheric relevance of the experimental results. This is particularly relevant because it increases the probability that these particles will encounter pollutant plumes, including those rich in glyoxal or its precursors, during their atmospheric lifetime.

In conclusion, while absolute concentrations differ, the experimental timescales, uptake dynamics, and size selection of aerosols all support the atmospheric relevance of our findings.

**Citation**: https://doi.org/10.5194/egusphere-2024-4073-RC2

Alvarado, L. M. A., Richter, A., Vrekoussis, M., Hilboll, A., Kalisz Hedegaard, A. B., Schneising, O., and Burrows, J. P.: Unexpected long-range transport of glyoxal and formaldehyde observed from the Copernicus Sentinel-5 Precursor satellite during the 2018 Canadian wildfires, Atmos. Chem. Phys., 20, 2057–2072, https://doi.org/10.5194/acp-20-2057-2020, 2020.

Fu, T.-M., Jacob, D. J., Wittrock, F., Burrows, J. P., Vrekoussis, M., and Henze, D. K.: Global budgets of atmospheric glyoxal and methylglyoxal, and implications for formation of secondary organic aerosols, J. Geophys. Res., 113, D15303, https://doi.org/10.1029/2007JD009505, 2008.

Battaglia, F., Baldo, C., Cazaunau, M., Bergé, A., Pangui, E., Picquet-Varrault, B., Doussin, J.-F., & Formenti, P. (2025). Protocol for generating realistic submicron mono-dispersed mineral dust particles in simulation chambers and laboratory experiments. Aerosol Science and Technology, 59(3), 357–369. https://doi.org/10.1080/02786826.2024.2442518

Myrdal, P. B. and Yalkowsky, S. H.: Estimating Pure Component Vapor Pressures of Complex Organic Molecules, Ind. Eng. Chem. Res., 36, 2494–2499, https://doi.org/10.1021/ie950242l, 1997.

Raventos-Duran, T., Camredon, M., Valorso, R., Mouchel-Vallon, C., and Aumont, B.: Structure-activity relationships to estimate the effective Henry's law constants of organics of atmospheric interest, Atmos. Chem. Phys., 10, 7643–7654, https://doi.org/10.5194/acp-10-7643-2010, 2010.

Shen, X., Wu, H., Zhao, Y., Huang, D., Huang, L., and Chen, Z.: Heterogeneous reactions of glyoxal on mineral particles: A new avenue for oligomers and organosulfate formation, Atmos. Environ., 131, 133–140, https://doi.org/10.1016/j.atmosenv.2016.01.048, 2016.

Volkamer, R., Spietz, P., Burrows, J., and Platt, U.: High-resolution absorption cross-section of glyoxal in the UV–vis and IR spectral ranges, Journal of Photochemistry and Photobiology A: Chemistry, 172, 35–46, https://doi.org/10.1016/j.jphotochem.2004.11.011, 2005.

Zhou, X. and Mopper, K.: Apparent partition coefficients of 15 carbonyl compounds between air and seawater and between air and freshwater; implications for air-sea exchange, Environ. Sci. Technol., 24, 1864–1869, https://doi.org/10.1021/es00082a013, 1990.

Zielinski, A. T., Kourtchev, I., Bortolini, C., Fuller, S. J., Giorio, C., Popoola, O. A. M., Bogialli, S., Tapparo, A., Jones, R. L., and Kalberer, M.: A new processing scheme for ultra-high resolution direct infusion mass spectrometry data, Atmos. Environ., 178, 129–139, https://doi.org/10.1016/j.atmosenv.2018.01.034, 2018.

Zogka, A. G., Lostier, A., Papadimitriou, V. C., Thevenet, F., Formenti, P., Rossi, M. J., Chen, H., and Romanias, M. N.: Unraveling the Uptake of Glyoxal on a Diversity of Natural Dusts and Surrogates: Linking Dust Composition to Glyoxal Uptake and Estimation of Atmospheric Lifetimes, ACS Earth Space Chem., 8, 1165–1178, https://doi.org/10.1021/acsearthspacechem.3c00359, 2024.

---

## Author Response (AR2)

**Reply to referee comments (RC) to manuscript egusphere-2024-4073**

The authors are grateful to anonymous reviewer 1 for providing additional feedbacks to the manuscript.

Our detailed point-by-point response to their comments are reported below. We believe that these modifications improved the reading and the understanding of the manuscript, fostering its significance.

**Reply to Referee # 1 comments (RC) to manuscript egusphere-2024-4073**

Dear authors, thanks for considering my comments. I have carefully reviewed the manuscript. This manuscript has been greatly improved. I recommend publication after addressing the points below.

We thank Referee #1 for the complete and thoughtful revision of our manuscript.

In the following the Referees' remarks and questions are in black and our replies are in light blue

1. Chemical Mechanisms and Oligomerization: The proposed reaction scheme (Fig 7, Page 27) is helpful but could be strengthened:

Suggestion: Explicitly discuss the role of the mineral surface in facilitating the proposed pathways (hydration, oxidation, oligomerization). Is it primarily providing an aqueous layer, acting as a catalyst (e.g., for acid-catalyzed oligomerization hinted at on Page 28), or participating directly in redox reactions? Link the observed product distribution (e.g., glycolic acid favored under light/O3) to potential surface-mediated reaction pathways. The detection of oligomers up to C10 is interesting. Discuss how the confinement on a particle surface vs. bulk aqueous phase might favor higher-order oligomerization (Page 28). Are the identified oligomer formulas consistent with mechanisms proposed for glyoxal in aqueous aerosols or on other surfaces? The absence of formic acid detection is noted (Page 23). Discuss potential reasons (e.g., rapid further oxidation, partitioning) in light of its common observation in other glyoxal uptake studies.

Following the reviewer's suggestion we have now expanded the discussion on how the mineral surface could play a role in the observed reactivity of glyoxal. At page 29 of the revised version, from line 725: "*Concerning hydration reactions, Shen et al. (2016) showed that both glyoxal and water can accumulate onto dust particle surfaces (as evidenced by FTIR spectra) promoting hydration reactions through surface confinement of the reagents. Absorption of water on dust surfaces can also promote oligomerisation both directly (hydration is the first step of the oligomerisation reaction) as well as indirectly by enhancing particles' ability to absorb glyoxal.*

*Concerning oxidation, leading to organic acid formation, the dust surface could contribute directly to this mechanism by providing redox reactive sites (Shen et al., 2016). Redox reactions promoted the formation of glycolic, glyoxylic, and oxalic acids.*"

Concerning formic acid, we hypothesize that we did not detect it in the particle phase due to the low amount of collected material on the filter which could mean that formic acid was below detection limit. In addition, the walls of the chamber provide a much larger surface compared to the dust which could make the partitioning of formic acid onto the dust less favourable. We have now amended the discussion at page 24, from line 605: "*Unlike previous studies (Galloway et al., 2009; Rubasinghege et al., 2013; Shen et al., 2016), formic acid was not detected on the particle phase despite high concentrations observed in the gas phase (Figure 2). It can be hypothesized that formic acid was not detected in the particle phase due to the low amount of organic material on the filter samples and the unsuitability of the method used (due to interferences at the retention time where formic acid is expected).*"

2. High Glyoxal Concentrations: The nominal initial glyoxal concentration (~1 ppmv) and measured peaks (370-1050 ppbv, Table 1) are significantly higher than typical ambient levels (ppt-ppbv). While the authors argue the first-order kinetics make γ transferable (Page 31), and Shen et al. showed decreasing γ with concentration, the potential for saturation effects or surface aging impacting the initial uptake kinetics observed here should be discussed. Suggestion: Explicitly discuss the potential impact of using such high concentrations on the measured initial uptake kinetics and product distribution. Can the results be confidently extrapolated to lower, atmospherically relevant concentrations? Consider citing Trainic et al. (2011) or similar regarding concentration effects on glyoxal uptake.

Referee is right about the possibility of surface saturation and this is what we observe and explicitly comment for figure 7. Trainic et al. (2011) compares the obtained mass growth rate and the ratio of final mass to seed mass (normalized by sulphate mass) and found them comparable to the previous results of Liggio et al. (2005) but higher than reported by Galloway et al (2009). Trainic et al. (2011) attribute these differences to differences in the initial seed sizes rather that in the initial glyoxal concentrations (which are a factor of 30 higher in Galloway et al. (2009)). Furthermore, we compared extensively the products of reactions, including for the control experiments with ammonium sulphate presented in the supplementary material (Text S5), and found in agreement with those studies.

Henceforth, in conclusion, we have now added an additional sentence in the conclusive remark section (lines 786-788) stating the following "After this time, the dust surface seems saturated, likely because of the excessive glyoxal concentrations injected in the chamber."

3. Page 7, Line 203: Typo: "REI" should be "RIE" (Relative Ionization Efficiency).

This has been corrected, thank you

4. Page 18, Lines 482-483: "The presence of ozone appears uninfluential." This is an important observation given the complex chemistry possible. Briefly speculate why ozone might not significantly influence glyoxal uptake or product formation on this specific dust.

The observation that ozone is uninfluential to heterogeneous reactions on mineral dust is not new. Hanish and Crowley (2003a) investigated the combined uptake of $O_3$ and $HNO_3$ onto dust, and showed that the uptake of $HNO_3$ on dust was not influenced by the presence of $O_3$ (and conversely, the uptake of $O_3$ was not influenced by the presence of $HNO_3$). These authors attributed these observations to the fact that in their experiments, $O_3$ concentrations were in excess by more than three orders of magnitude with respect to those of $HNO_3$, compensating for the fact that the uptake coefficients for $O_3$ is approximately four orders of magnitude lower than for $HNO_3$ (Hanish and Croley 2003b; Chang et al., 2005). In these conditions, which are not generally met in the atmosphere, the presence of $O_3$ could result in the modification of surface chemical characteristics, or to competition for adsorption sites of the two species. These considerations are applicable to our experiments, as the uptake coefficients of $O_3$ and glyoxal are of comparable magnitudes and we use comparable concentrations. These considerations have been added to the main manuscript at line 485.

References

Chang, R. Y.-W., Sullivan, R. C., and Abbatt, J. P. D.: Initial uptake of ozone on Saharan dust at atmospheric relative humidities, Geophysical Research Letters, 32, https://doi.org/10.1029/2005GL023317, 2005.

Galloway, M. M., Chhabra, P. S., Chan, A. W. H., Surratt, J. D., Flagan, R. C., Seinfeld, J. H., and Keutsch, F. N.: Glyoxal uptake on ammonium sulphate seed aerosol: reaction products and reversibility of uptake under dark and irradiated conditions, Atmos. Chem. Phys., 2009.

Hanisch, F., and Crowley, J. N.: Heterogeneous reactivity of NO and HNO3 on mineral dust in the presence of ozone, Physical Chemistry Chemical Physics, 5, 883-887, 10.1039/b211503d, 2003a.

Hanisch, F. and Crowley, J. N.: Ozone decomposition on Saharan dust: an experimental investigation, Atmos. Chem. Phys., 3, 119–130, https://doi.org/10.5194/acp-3-119-2003, 2003b.

Liggio, J.: Reactive uptake of glyoxal by particulate matter, J. Geophys. Res., 110, D10304, https://doi.org/10.1029/2004JD005113, 2005a.

Rubasinghege, G., Ogden, S., Baltrusaitis, J., and Grassian, V. H.: Heterogeneous Uptake and Adsorption of Gas-Phase Formic Acid on Oxide and Clay Particle Surfaces: The Roles of Surface Hydroxyl Groups and Adsorbed Water in Formic Acid Adsorption and the Impact of Formic Acid Adsorption on Water Uptake, J. Phys. Chem. A, 117, 11316–11327, https://doi.org/10.1021/jp408169w, 2013.

Shen, X., Wu, H., Zhao, Y., Huang, D., Huang, L., and Chen, Z.: Heterogeneous reactions of glyoxal on mineral particles: A new avenue for oligomers and organosulfate formation, Atmospheric Environment, 131, 133–140, https://doi.org/10.1016/j.atmosenv.2016.01.048, 2016.

Trainic, M., Abo Riziq, A., Lavi, A., Flores, J. M., and Rudich, Y.: The optical, physical and chemical properties of the products of glyoxal uptake on ammonium sulfate seed

aerosols, Atmos. Chem. Phys., 11, 9697–9707, https://doi.org/10.5194/acp-11-9697-2011, 2011.

---

## Author Response (AR3)

Créteil, France – 22 July 2025

Dear Editor,

Please find hereby the revised version of manuscript egusphere-2024-4073 submitted in December 2024 to Atmos Chem Phys.

We have tried to address as best as possible the additional comments of Referee #1, whom we would like to thank sincerely.

Please note

1. Label of figure 5 is corrected

2. Figures supplements are corrected

3. References have been modify to follow journal criteria.

4. Formatting has been checked

We hope that this research paper will retain your attention for publication on Atmospheric Chemistry and Physics

With my very best regards

Paola Formenti, corresponding author